# ROBUST SUBSPACE RECOVERY LAYER FOR UNSUPERVISED ANOMALY DETECTION

**Chieh-Hsin Lai,**\* **Dongmian Zou**\* **& Gilad Lerman**
School of Mathematics
University of Minnesota
Minneapolis, MN 55455
{laixx313, dzou, lerman}@umn.edu

## ABSTRACT

We propose a neural network for unsupervised anomaly detection with a novel robust subspace recovery layer (RSR layer). This layer seeks to extract the underlying subspace from a latent representation of the given data and removes outliers that lie away from this subspace. It is used within an autoencoder. The encoder maps the data into a latent space, from which the RSR layer extracts the subspace. The decoder then smoothly maps back the underlying subspace to a "manifold" close to the original inliers. Inliers and outliers are distinguished according to the distances between the original and mapped positions (small for inliers and large for outliers). Extensive numerical experiments with both image and document datasets demonstrate state-of-the-art precision and recall.

## 1 INTRODUCTION

Finding and utilizing patterns in data is a common task for modern machine learning systems. However, there is often some anomalous information that does not follow a common pattern and has to be recognized. For this purpose, anomaly detection aims to identify data points that "do not conform to expected behavior" (Chandola et al., 2009). We refer to such points as either anomalous or outliers. In many applications, there is no ground truth available to distinguish anomalous from normal points, and they need to be detected in an unsupervised fashion. For example, one may need to remove anomalous images from a set of images obtained by a search engine without any prior knowledge about how a normal image should look (Xia et al., 2015). Similarly, one may need to distinguish unusual news items from a large collection of news documents without any information whether a news item is usual or not (Kannan et al., 2017). In these examples, the only assumptions are that normal data points appear more often than anomalous ones and have a simple underlying structure which is unknown to the user.

Some early methods for anomaly detection relied on Principal Component Analysis (PCA) (Shyu et al., 2003). Here one assumes that the underlying unknown structure of the normal samples is linear. However, PCA is sensitive to outliers and will often not succeed in recovering the linear structure or identifying the outliers (Lerman & Maunu, 2018; Vaswani & Narayanamurthy, 2018). More recent ideas of Robust PCA (RPCA) (Wright et al., 2009; Vaswani & Narayanamurthy, 2018) have been considered for some specific problems of anomaly detection or removal (Zhou & Paffenroth, 2017; Paffenroth et al., 2018). RPCA assumes sparse corruption, that is, few elements of the data matrix are corrupted. This assumption is natural for some special problems in computer vision, in particular, background subtraction (De La Torre & Black, 2003; Wright et al., 2009; Vaswani & Narayanamurthy, 2018). However, a natural setting of anomaly detection with hidden linear structure may assume instead that a large portion of the data points are fully corrupted. The mathematical framework that addresses this setting is referred to as robust subspace recovery (RSR) (Lerman & Maunu, 2018).

While Robust PCA and RSR try to extract linear structure or identify outliers lying away from such structure, the underlying geometric structure of many real datasets is nonlinear. Therefore, one

---

\*Equal contribution.

needs to extract crucial features of the nonlinear structure of the data while being robust to outliers. In order to achieve this goal, we propose to use an autoencoder (composed of an encoder and a decoder) with an RSR layer. We refer to it as RSRAE (RSR autoencoder). It aims to robustly and nonlinearly reduce the dimension of the data in the following way. The encoder maps the data into a high-dimensional space. The RSR layer linearly maps the embedded points into a low-dimensional subspace that aims to learn the hidden linear structure of the embedded normal points. The decoder maps the points from this subspace to the original space. It aims to map the normal points near their original locations, and the anomalous points far from their original locations.

Ideally, the encoder maps the normal data to a linear space and any anomalies lie away from this subspace. In this ideal scenario, anomalies can be removed by an RSR method directly applied to the data embedded by the encoder. Since the linear model for the normal data embedded by the encoder is only approximate, we do not directly apply RSR to the embedded data. Instead, we minimize a sum of the reconstruction error of the autoencoder and the RSR error for the data embedded by the encoder. We advocate for an alternating procedure, so that the parameters of the autoencoder and the RSR layer are optimized in turn.

## 1.1 Structure of the rest of the paper

Section 2 reviews works that are directly related to the proposed RSRAE and highlights the original contributions of this paper. Section 3 explains the proposed RSRAE, and in particular, its RSR layer and total energy function. Section 4 includes extensive experimental evidence demonstrating effectiveness of RSRAE with both image and document data. Section 5 discusses theory for the relationship of the RSR penalty with the WGAN penalty. Section 6 summarizes this work and mentions future directions.

## 2 Related Works and Contribution

We review related works in Section 2.1 and highlight our contribution in Section 2.2.

## 2.1 Related Works

Several recent works have used autoencoders for anomaly detection. Xia et al. (2015) proposed the earliest work on anomaly detection via an autoencoder, while utilizing large reconstruction error of outliers. They apply an iterative and cyclic scheme, where in each iteration, they determine the inliers and use them for updating the parameters of the autoencoder. Aytekin et al. (2018) apply $\ell_2$ normalization for the latent code of the autoencoder and also consider the case of multiple modes for the normal samples. Instead of using the reconstruction error, they apply $k$-means clustering for the latent code, and identify outliers as points whose latent representations are far from all the cluster centers. Zong et al. (2018) also use an autoencoder with clustered latent code, but they fit a Gaussian Mixture Model using an additional neural network. Restricted Boltzmann Machines (RBMs) are similar to autoencoders. Zhai et al. (2016) define "energy functions" for RBMs that are similar to the reconstruction losses for autoencoders. They identify anomalous samples according to large energy values. Chalapathy et al. (2017) propose using ideas of RPCA within an autoencoder, where they alternatively optimize the parameters of the autoencoder and a sparse residual matrix.

The above works are designed for datasets with a small fraction of outliers. However, when this fraction increases, outliers are often not distinguished by high reconstruction errors or low similarity scores. In order to identify them, additional assumptions on the structure of the normal data need to be incorporated. For example, Zhou & Paffenroth (2017) decompose the input data into two parts: low-rank and sparse (or column-sparse). The low-rank part is fed into an autoencoder and the sparse part is imposed as a penalty term with the $\ell_1$-norm (or $\ell_{2,1}$-norm for column-sparsity).

In this work, we use a term analogous to the $\ell_{2,1}$-norm, which can be interpreted as the sum of absolute deviations from a latent subspace. However, we do not decompose the data a priori, but minimize an energy combining this term and the reconstruction error. Minimization of the former term is known as least absolute deviations in RSR (Lerman & Maunu, 2018). It was first suggested for RSR and related problems in Watson (2001); Ding et al. (2006); Zhang et al. (2009). The robustness to outliers of this energy, or of relaxed versions of it, was studied in McCoy & Tropp (2011); Xu

et al. (2012); Lerman & Zhang (2014); Zhang & Lerman (2014); Lerman et al. (2015); Lerman & Maunu (2017); Maunu et al. (2017). In particular, Maunu et al. (2017) established its well-behaved landscape under special, though natural, deterministic conditions. Under similar conditions, they guaranteed fast subspace recovery by a simple algorithm that aims to minimize this energy.

Another directly related idea for extracting useful latent features is an addition of a linear self-expressive layer to an autoencoder (Ji et al., 2017). It is used in the different setting of unsupervised subspace clustering. By imposing the self-expressiveness, the autoencoder is robust to an increasing number of clusters. Although self-expressiveness also improves robustness to noise and outliers, Ji et al. (2017) aims at clustering and thus its goal is different than ours. Furthermore, their self-expressive energy does not explicitly consider robustness, while ours does. Lezama et al. (2018) consider a somewhat parallel idea of imposing a loss function to increase the robustness of representation. However, their goal is to increase the margin between classes and their method only applies to a supervised setting in anomaly detection, where the normal data is multi-modal.

## 2.2 CONTRIBUTION OF THIS WORK

This work introduces an RSR layer within an autoencoder. It incorporates a special regularizer that enforces an outliers-robust linear structure in the embedding obtained by the encoder. We clarify that the method does not alternate between application of the autoencoder and the RSR layer, but fully integrates these two components. Our experiments demonstrate that a simple incorporation of a "robust loss" within a regular autoencoder does not work well for anomaly detection. We try to explain this and also the improvement obtained by incorporating an additional RSR layer.

Our proposed architecture is simple to implement. Furthermore, the RSR layer is not limited to a specific design of RSRAE but can be put into any well-designed autoencoder structure. The epoch time of the proposed algorithm is comparable to those of other common autoencoders. Furthermore, our experiments show that RSRAE competitively performs in unsupervised anomaly detection tasks.

RSRAE addresses the unsupervised setting, but is not designed to be highly competitive in the semi-supervised or supervised settings, where one has access to training data from the normal class or from both classes, respectively. In these settings, RSRAE functions like a regular autoencoder without taking an advantage of its RSR layer, unless the training data for the normal class is corrupted with outliers.

The use of RSR is not restricted to autoencoders. We establish some preliminary analysis for RSR within a generative adversarial network (GAN) (Goodfellow et al., 2014; Arjovsky et al., 2017) in Section 5. More precisely, we show that a linear WGAN intrinsically incorporates RSR in some special settings, although it is unclear how to impose an RSR layer.

## 3 RSR LAYER FOR OUTLIER REMOVAL

We assume input data $\{\mathbf{x}^{(t)}\}_{t=1}^N$ in $\mathbb{R}^M$, and denote by $\mathbf{X}$ its corresponding data matrix, whose $t$-th column is $\mathbf{x}^{(t)}$. The encoder of RSRAE, $\mathscr{E}$, is a neural network that maps each data point, $\mathbf{x}^{(t)}$, to its latent code $\mathbf{z}^{(t)} = \mathscr{E}(\mathbf{x}^{(t)}) \in \mathbb{R}^D$. The RSR layer is a linear transformation $\mathbf{A} \in \mathbb{R}^{d \times D}$ that reduces the dimension to $d$. That is, $\tilde{\mathbf{z}}^{(t)} = \mathbf{A}\mathbf{z}^{(t)} \in \mathbb{R}^d$. The decoder $\mathscr{D}$ is a neural network that maps $\tilde{\mathbf{z}}^{(t)}$ to $\tilde{\mathbf{x}}^{(t)}$ in the original ambient space $\mathbb{R}^M$.

We can write the forward maps in a compact form using the corresponding data matrices as follows:

$$\mathbf{Z} = \mathscr{E}(\mathbf{X}), \quad \tilde{\mathbf{Z}} = \mathbf{A}\mathbf{Z}, \quad \tilde{\mathbf{X}} = \mathscr{D}(\tilde{\mathbf{Z}}). \tag{1}$$

Ideally, we would like to optimize RSRAE so it only maintains the underlying structure of the normal data. We assume that the original normal data lies on a $d$-dimensional "manifold" in $\mathbb{R}^D$ and thus the RSR layer embeds its latent code into $\mathbb{R}^d$. In this ideal optimization setting, the similarity between the input and the output of RSRAE is large whenever the input is normal and small whenever the input is anomalous. Therefore, by thresholding a similarity measure, one may distinguish between normal and anomalous data points.

In practice, the matrix $\mathbf{A}$ and the parameters of $\mathscr{E}$ and $\mathscr{D}$ are obtained by minimizing a loss function, which is a sum of two parts: the reconstruction loss from the autoencoder and the loss from the RSR

layer. For $p > 0$, an $\ell_{2,p}$ reconstruction loss for the autoencoder is

$$L_{\mathrm{AE}}^p(\mathscr{E}, \mathbf{A}, \mathscr{D}) = \sum_{t=1}^N \left\| \mathbf{x}^{(t)} - \tilde{\mathbf{x}}^{(t)} \right\|_2^p . \tag{2}$$

In order to motivate our choice of RSR loss, we review a common formulation for the original RSR problem. In this problem one needs to recover a linear subspace, or equivalently an orthogonal projection $\mathbf{P}$ onto this subspace. Assume a dataset $\{\mathbf{y}^{(t)}\}_{t=1}^N$ and let $\mathbf{I}$ denote the identity matrix in the ambient space of the dataset. The goal is to find an orthogonal projector $\mathbf{P}$ of dimension $d$ whose subspace robustly approximates this dataset. The least $q$-th power deviations formulation for $q > 0$, or least absolute deviations when $q = 1$ (Lerman & Maunu, 2018), seeks $\mathbf{P}$ that minimizes

$$\hat{L}(\mathbf{P}) = \sum_{t=1}^N \left\| (\mathbf{I} - \mathbf{P}) \, \mathbf{y}^{(t)} \right\|_2^q . \tag{3}$$

The solution of this problem is robust to some outliers when $q \leq 1$ (Lerman & Zhang, 2014; Lerman & Maunu, 2017); furthermore, $q < 1$ can result in a wealth of local minima and thus $q = 1$ is preferable (Lerman & Zhang, 2014; Lerman & Maunu, 2017).

A similar loss function to (3) for RSRAE is

$$\begin{aligned} L_{\mathrm{RSR}}^q(\mathbf{A}) &= \lambda_1 L_{\mathrm{RSR}_1}(\mathbf{A}) + \lambda_2 L_{\mathrm{RSR}_2}(\mathbf{A}) \\ &:= \lambda_1 \sum_{t=1}^N \left\| \mathbf{z}^{(t)} - \mathbf{A}^{\mathrm{T}} \underbrace{\mathbf{A} \mathbf{z}^{(t)}}_{\tilde{\mathbf{z}}^{(t)}} \right\|_2^q + \lambda_2 \left\| \mathbf{A} \mathbf{A}^{\mathrm{T}} - \mathbf{I}_d \right\|_{\mathrm{F}}^2 , \end{aligned} \tag{4}$$

where $\mathbf{A}^{\mathrm{T}}$ denotes the transpose of $\mathbf{A}$, $\mathbf{I}_d$ denotes the $d \times d$ identity matrix and $\|\cdot\|_{\mathrm{F}}$ denotes the Frobenius norm. Here $\lambda_1, \lambda_2 > 0$ are predetermined hyperparameters, though we later show that one may solve the underlying problem without using them. We note that the first term in the weighted sum of (4) is close to (3) as long as $\mathbf{A}^{\mathrm{T}} \mathbf{A}$ is close to an orthogonal projector. To enforce this requirement we introduced the second term in the weighted sum of (4). In Appendix C we discuss further properties of the RSR energy and its minimization.

To emphasize the effect of outlier removal, we take $p = 1$ in (2) and $q = 1$ in (4). That is, we use the $l_{2,1}$ norm, or the formulation of least absolute deviations, for both reconstruction and RSR. The loss function of RSRAE is the sum of the two loss terms in (2) and (4), that is,

$$L_{\mathrm{RSRAE}}(\mathscr{E}, \mathbf{A}, \mathscr{D}) = L_{\mathrm{AE}}^1(\mathscr{E}, \mathbf{A}, \mathscr{D}) + L_{\mathrm{RSR}}^1(\mathbf{A}). \tag{5}$$

We remark that the sole minimization of $L_{\mathrm{AE}}^1$, without $L_{\mathrm{RSR}}^1$, is not effective for anomaly detection. We numerically demonstrate this in Section 4.3 and also try to explain it in Section 5.1.

Our proposed algorithm for optimizing (5), which we refer to as the RSRAE algorithm, uses alternating minimization. It iteratively backpropagates the three terms $L_{\mathrm{AE}}^1$, $L_{\mathrm{RSR}_1}$, $L_{\mathrm{RSR}_2}$ and accordingly updates the parameters of the RSR autoencoder. For clarity, we describe this basic procedure in Algorithm 1 of Appendix A. It is independent of the values of the parameters $\lambda_1$ and $\lambda_2$. Note that the additional gradient step with respect to the RSR loss just updates the parameters in $\mathbf{A}$. Therefore it does not significantly increase the epoch time of a standard autoencoder for anomaly detection. Another possible method, which we refer to as RSRAE+, is direct minimization of $L_{\mathrm{RSRAE}}$ with predetermined $\lambda_1$ and $\lambda_2$ via auto-differentiation (see Algorithm 2 of Appendix A). Section 4.3 and Appendix I.2 demonstrate that in general, RSRAE performs better than RSRAE+, though it is possible that similar performance can be achieved by carefully tuning the parameters $\lambda_1$ and $\lambda_2$ when implementing RSRAE+.

We remark that a standard autoencoder is obtained by minimizing only $L_{\mathrm{AE}}^2$, without the RSR loss. One might hope that minimizing $L_{\mathrm{AE}}^1$ may introduce the needed robustness. However, Section 4.3 and Appendix I.2 demonstrate that results obtained by minimizing $L_{\mathrm{AE}}^1$ or $L_{\mathrm{AE}}^2$ are comparable, and are worse than those of RSRAE and RSRAE+.

## 4 EXPERIMENTAL RESULTS

We test our method [1] on five datasets: Caltech 101 (Fei-Fei et al., 2007), Fashion-MNIST (Xiao et al., 2017), Tiny Imagenet (a small subset of Imagenet (Russakovsky et al., 2015)), Reuters-21578 (Lewis, 1997) and 20 Newsgroups (Lang, 1995).

**Caltech 101** contains 9,146 RGB images labeled according to 101 distinct object categories. We take the 11 categories that contain at least 100 images and randomly choose 100 images per category. We preprocess all 1100 images to have size $32 \times 32 \times 3$ and pixel values normalized between $-1$ and $1$. In each experiment, the inliers are the 100 images from a certain category and we sample $c \times 100$ outliers from the rest of 1000 images of other categories, where $c \in \{0.1, 0.3, 0.5, 0.7, 0.9\}$.

**Fashion-MNIST** contains $28 \times 28$ grayscale images of clothing and accessories, which are categorized into 10 classes. We use the test set which contains 10,000 images and normalize pixel values to lie in $[-1, 1]$. In each experiment, we fix a class and the inliers are the test images in this class. We randomly sample $c \times 1,000$ outliers from the rest of classes (here and below $c$ is as above). Since there are around 1000 test images in each class, the outlier ratio is approximately $c$.

**Tiny Imagenet** contains 200 classes of RGB images from a distinct subset of Imagenet. We select 10 classes with 500 training images per class. We preprocess the images to have size $32 \times 32 \times 3$ and pixel values in $[-1, 1]$. We further represent the images by deep features obtained by a ResNet (He et al., 2016) with dimension 256 (Appendix I.1 provides results for the raw images). In each experiment, 500 inliers are from a fixed class and $c \times 500$ outliers are from the rest of classes.

**Reuters-21578** contains 90 text categories with multi-labels. We consider the five largest classes with single labels and randomly sample from them 360 documents per class. The documents are preprocessed into vectors of size 26,147 by sequentially applying the TFIDF transformer and Hashing vectorizer (Rajaraman & Ullman, 2011). In each experiment, the inliers are the documents of a fixed class and $c \times 360$ outliers are randomly sampled from the other classes.

**20 Newsgroups** contains newsgroup documents with 20 different labels. We sample 360 documents per class and preprocess them as above into vectors of size 10,000. In each experiment, the inliers are the documents from a fixed class and $c \times 360$ outliers are sampled from the other classes.

### 4.1 BENCHMARKS AND SETTING

We compare RSRAE with the following benchmarks: Local Outlier Factor (LOF) (Breunig et al., 2000), One-Class SVM (OCSVM) (Schölkopf et al., 2000; Amer et al., 2013), Isolation Forest (IF) (Liu et al., 2012), Deep Structured Energy Based Models (DSEBMs) (Zhai et al., 2016), Geometric Transformations (GT) (Golan & El-Yaniv, 2018), and Deep Autoencoding Gaussian Mixture Model (DAGMM) (Zong et al., 2018). Of those benchmarks, LOF, OCSVM and IF are traditional, while powerful methods, for unsupervised anomaly detection and do not involve neural networks. DSEBMs, DAGMM and GT are more recent and all involve neural networks. DSEBMs is built for unsupervised anomaly detection. DAGMM and GT are designed for semi-supervised anomaly detection, but allow corruption. We use them to learn a model for the inliers and assign anomaly scores using the combined set of both inliers and outliers. GT only applies to image data. We briefly describe these methods in Appendix E.

We implemented DSEBMs, DAGMM and GT using the codes [2] from Golan & El-Yaniv (2018) with minimal modification so that they adapt to the data described above and the available GPUs in our machine. The LOF, OCSVM and IF methods are adapted from the scikit-learn packages.

We describe the structure of the RSRAE as follows. For the image datasets without deep features, the encoder consists of three convolutional layers: $5 \times 5$ kernels with 32 output channels, strides 2; $5 \times 5$ kernels with 64 output channels, strides 2; and $3 \times 3$ kernels with 128 output channels, strides 2. The output of the encoder is flattened and the RSR layer transforms it into a 10-dimensional vector. That is, we fix $d = 10$ in all experiments. The decoder consists of a dense layer that maps the output of the RSR layer into a vector of the same shape as the output of the encoder, and three deconvolutional layers: $3 \times 3$ kernels with 64 output channels, strides 2; $5 \times 5$ kernels with 32

---

[1] Our implementation is available at https://github.com/dmzou/RSRAE.git

[2] https://github.com/izikgo/AnomalyDetectionTransformations

output channels, strides 2; $5 \times 5$ kernels with 1 (grayscale) or 3 (RGB) output channels, strides 2. For the preprocessed document datasets or the deep features of Tiny Imagenet, the encoder is a fully connected network with size (32, 64, 128), the RSR layer linearly maps the output of the encoder to dimension 10, and the decoder is a fully connected network with size (128, 64, 32, $D$) where $D$ is the dimension of the input. Batch normalization is applied to each layer of the encoders and the decoders. The output of the RSR layer is $\ell_2$-normalized before applying the decoder. For DSEBMs and DAGMM we use the same number of layers and the same dimensions in each layer for the autoencoder as in RSRAE. For each experiment, the RSRAE model is optimized with Adam using a learning rate of 0.00025 and 200 epochs. The batch size is 128 for each gradient step. The setting of training is consistent for all the neural network based methods.

The two main hyperparameters of RSRAE are the intrinsic dimension $d$ and learning rate. Their values were fixed above. Appendix G demonstrates stability to changes in these values.

All experiments were executed on a Linux machine with 64GB RAM and four GTX1080Ti GPUs. For all experiments with neural networks, we used TensorFlow and Keras. We report runtimes in Appendix H.

### 4.2 RESULTS

We summarize the precision and recall of our experiments by the AUC (area under curve) and AP (average precision) scores. For completeness, we include the definitions of these common scores in Appendix E. We compute them by considering the outliers as "positive". We remark that we did not record the precision-recall-F1 scores, as in Xia et al. (2015); Zong et al. (2018), since in practice it requires knowledge of the outlier ratio.

Figs. 1 and 2 present the AUC and AP scores of RSRAE and the methods described in Section 4.1 for the datasets described above, where GT is only applied to image data without deep features. For each constant $c$ (the outlier ratio) and each method, we average the AUC and AP scores over 5 runs with different random initializations and also compute the standard deviations. For brevity of presentation, we report the averaged scores among all classes and designate the averaged standard deviations by bars.

The results indicates that RSRAE clearly outperforms other methods in most cases, especially when $c$ is large. Indeed, the RSR layer was designed to handle large outlier ratios. For Fashion MNIST and Tiny Imagenet with deep features, IF performs similarly to RSRAE, but IF performs poorly on the document datasets. OCSVM is the closest to RSRAE for the document datasets but it is generally not so competitive for the image datasets.

### 4.3 COMPARISON WITH VARIATIONS OF RSRAE

We use one image dataset (Caltech 101) and one document dataset (Reuters-21578) and compare between RSRAE and three variations of it. The first one is RSRAE+ (see Section 3) with $\lambda_1 = \lambda_2 = 0.1$ in (4) (these parameters were optimized on 20 Newsgroup, though results with other choices of parameters are later demonstrated in Section G.3). The next two are simpler autoencoders without RSR layers: AE-1 minimizes $L_{\mathrm{AE}}^1$, the $\ell_{2,1}$ reconstruction loss; and AE minimizes $L_{\mathrm{AE}}^2$, the $\ell_{2,2}$ reconstruction loss (it is a regular autoencoder for anomaly detection). We maintain the same architecture as that of RSRAE, including the matrix $\mathbf{A}$, but use different loss functions.

Fig. 3 reports the AUC and AP scores. We see that for the two datasets RSRAE+ with the prespecified $\lambda_1$ and $\lambda_2$ does not perform as well as RSRAE, but its performance is still better than AE and AE-1. This is expected since we chose $\lambda_1$ and $\lambda_2$ after few trials with a different dataset, whereas RSRAE is independent of these parameters. The performance of AE and AE-1 is clearly worse, and they are also not as good as some methods compared with in Section 4.2. At last, AE is generally comparable with AE-1. Similar results are noticed for the other datasets in Appendix I.2.

## 5 RELATED THEORY FOR THE RSR PENALTY

We explain here why we find it natural to incorporate RSR within a neural network. In Section 5.1 we first review the mathematical idea of an autoencoder and discuss the robustness of a linear

autoencoder with an $\ell_{2,1}$ loss (i.e., RSR loss). We then explain why a general autoencoder with an $\ell_{2,1}$ loss is not expected to be robust to outliers and why an RSR layer can improve its robustness. Section 5.2 is a first step of extending this view to a generative network. It establishes some robustness of WGAN with a linear generator, but the extension of an RSR layer to WGAN is left as an open problem.

### 5.1 ROBUSTNESS AND RELATED PROPERTIES OF AUTOENCODERS

Mathematically, an autoencoder for a dataset $\{\mathbf{x}^{(t)}\}_{t=1}^{N} \subset \mathbb{R}^D$ and a latent dimension $d < D$ is composed of an encoder $\mathscr{E} : \mathbb{R}^D \to \mathbb{R}^d$ and a decoder $\mathscr{D} : \mathbb{R}^d \to \mathbb{R}^D$ that minimize the following energy function with $p = 2$:

$$\sum_{t=1}^{N} \left\| \mathbf{x}^{(t)} - \mathscr{D} \circ \mathscr{E}(\mathbf{x}^{(t)}) \right\|_2^p , \tag{6}$$

where $\circ$ denotes function decomposition. It is a natural nonlinear generalization of PCA (Goodfellow et al., 2016). Indeed, in the case of a linear autoencoder, $\mathscr{E}$ and $\mathscr{D}$ are linear maps represented by matrices $\mathbf{E} \in \mathbb{R}^{d \times D}$ and $\mathbf{D} \in \mathbb{R}^{D \times d}$, respectively, that need to minimize (among such matrices) the following loss function with $p = 2$

$$\sum_{t=1}^{N} \left\| \mathbf{x}^{(t)} - \mathbf{D}\mathbf{E}\mathbf{x}^{(t)} \right\|_2^p . \tag{7}$$

We explain in Appendix D.1 that if $(\mathbf{D}^\star, \mathbf{E}^\star)$ is a minimizer of (7) with $p = 2$ (among $\mathbf{E} \in \mathbb{R}^{d \times D}$ and $\mathbf{D} \in \mathbb{R}^{D \times d}$), then $\mathbf{D}^\star\mathbf{E}^\star$ is the orthoprojector on the $d$-dimensional PCA subspace. This means, that the latent code $\{\mathbf{E}^\star\mathbf{x}^{(t)}\}_{t=1}^{N}$ parametrizes the PCA subspace and an additional application of $\mathbf{D}^\star$ to $\{\mathbf{E}^\star\mathbf{x}^{(t)}\}_{t=1}^{N}$ results in the projections of the data points $\{\mathbf{x}^{(t)}\}_{t=1}^{N}$ onto the PCA subspace. The recovery error for data points on this subspace is zero (as $\mathbf{D}^\star\mathbf{E}^\star$ is the identity on this subspace), and in general, this error is the Euclidean distance to the PCA subspace, $\left\| \mathbf{x}^{(t)} - \mathbf{D}^\star\mathbf{E}^\star\mathbf{x}^{(t)} \right\|_2$.

Intuitively, the idea of a general autoencoder is the same. It aims to fit a nice structure, such as a manifold, to the data, where ideally $\mathscr{D} \circ \mathscr{E}$ is a projection onto this nice structure. This idea can only be made rigorous for data approximated by simple geometric structure, e.g., by a graph of a sufficiently smooth function.

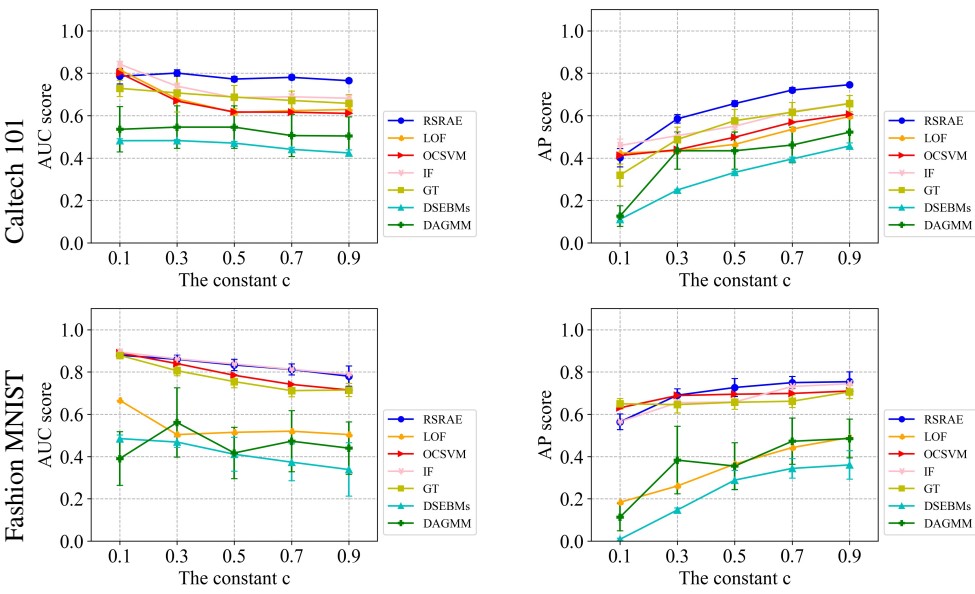

Figure 1: AUC and AP scores for RSRAE using Caltech 101 and Fashion MNIST.

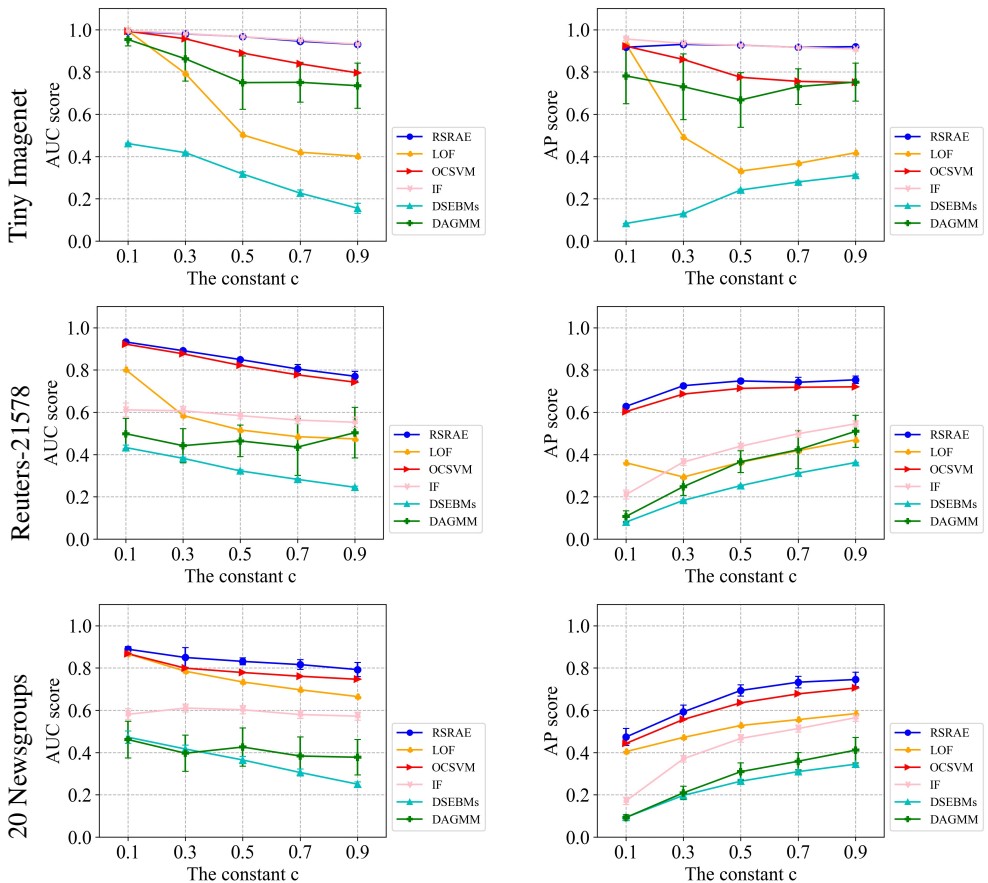

Figure 2: AUC and AP scores for RSRAE using Tiny Imagenet with deep features, Reuters-21578 and 20 Newsgroups.

In order to extend these methods to anomaly detection, one needs to incorporate robust strategies, so that the methods can still recover the underlying structure of the inliers, and consequently assign lower recovery errors for the inliers and higher recovery errors for the outliers. For example, in the linear case, one may assume a set of inliers lying on and around a subspace and an arbitrary set of outliers (with some restriction on their fraction). PCA, and equivalently, the linear autoencoder that minimizes (7) with $p = 2$, is not robust to general outliers. Thus it is not expected to distinguish well between inliers and outliers in this setting. As explained in Appendix D.1, minimizing (7) with $p = 1$ gives rise to the least absolute deviations subspace. This subspace can be robust to outliers under some conditions, but these conditions are restrictive (see examples in Lerman & Zhang (2014)). In order to deal with more adversarial outliers, it is advised to first normalize the data to the sphere (after appropriate centering) and then estimate the least absolute deviations subspace. This procedure was theoretically justified for a general setting of adversarial outliers in Maunu & Lerman (2019).

As in the linear case, an autoencoder that uses the loss function in (6) with $p = 1$ may not be robust to adversarial outliers. Unlike the linear case, there are no simple normalizations for this case. Indeed, the normalization to the sphere can completely distort the structure of an underlying manifold and it is also hard to center in this case. Furthermore, there are some obstacles of establishing robustness for the nonlinear case even under special assumptions.

Our basic idea for a robust autoencoder is to search for a latent low-dimensional code for the inliers within a larger embedding space. The additional RSR loss focuses on parametrizing the low-dimensional subspace of the encoded inliers, while being robust to outliers. Following the above discussion, we enhance such robustness by applying a normalization similar to the one discussed above, but adapted better to the structure of the network (see Section 4.1). The emphasis of the RSR

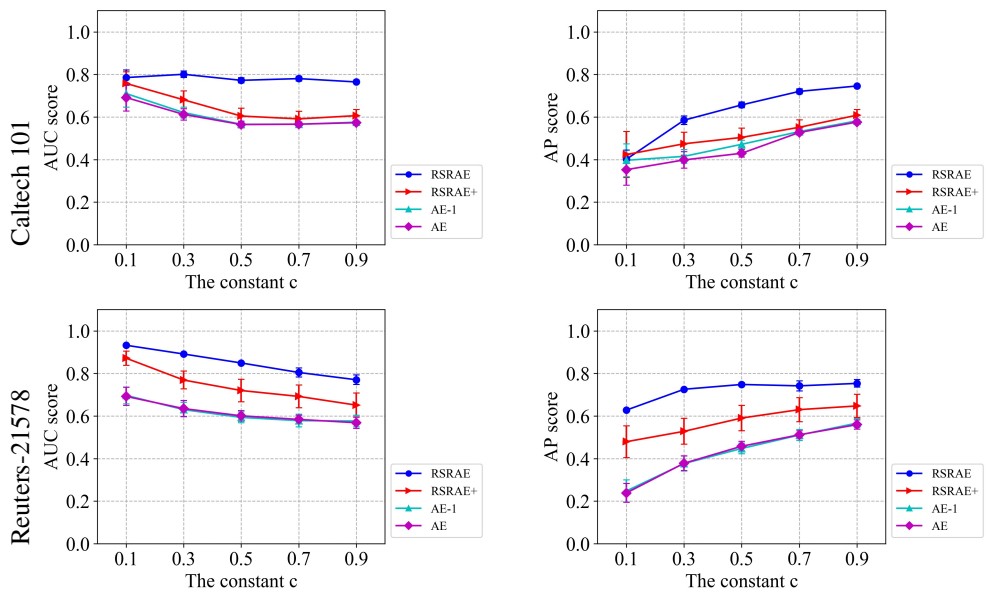

Figure 3: AUC and AP scores for RSRAE and alternative formulations using Caltech 101 and Reuters-21578.

layer is on appropriately encoding the inliers, where the encoding of the outliers does not matter. It is okay for the encoded outliers to lie within the subspace of the encoded inliers, as this will result in large recovery errors for the outliers. However, in general, most encoded outliers lie away from this subspace, and this is why such a mechanism is needed (otherwise, a regular autoencoder may obtain a good embedding).

## 5.2 RELATIONSHIP OF THE RSR LOSS WITH LINEARLY GENERATED WGAN

An open problem is whether RSR can be used within other neural network structures for unsupervised learning, such as variational autoencoders (VAEs) (Kingma & Welling, 2013) and generative adversarial networks (GANs) (Goodfellow et al., 2014). The latter two models are used in anomaly detection with a score function similar to the reconstruction error (An & Cho, 2015; Vasilev et al., 2018; Zenati et al., 2018; Kliger & Fleishman, 2018).

While we do not solve this problem, we establish a natural relationship between RSR and Wasserstein-GAN (WGAN) (Arjovsky et al., 2017; Gulrajani et al., 2017) with a linear generator, which is analogous to the example of a linear autoencoder mentioned above.

Let $W_p$ denote the $p$-Wasserstein distance in $\mathbb{R}^D$ ($p \geq 1$). That is, for two probability distributions $\mu, \nu$ on $\mathbb{R}^D$,

$$W_p(\mu, \nu) = \left( \inf_{\pi \in \Pi(\mu, \nu)} \mathbb{E}_{(\mathbf{x}, \mathbf{y}) \sim \pi} \|\mathbf{x} - \mathbf{y}\|_2^p \right)^{1/p}, \tag{8}$$

where $\Pi(\mu, \nu)$ is the set of joint distributions with $\mu, \nu$ as marginals. We formulate the following proposition (while prove it later in Appendix D.2) and then interpret it.

**Proposition 5.1.** *Let $p \geq 1$ and $\mu$ be a Gaussian distribution on $\mathbb{R}^D$ with mean $\mathbf{m}_X \in \mathbb{R}^D$ and full-rank covariance matrix $\mathbf{\Sigma}_X \in \mathbb{R}^{D \times D}$ (that is, $\mu$ is $\mathcal{N}(\mathbf{m}_X, \mathbf{\Sigma}_X)$). Then*

$$\min_{\nu \text{ is } \mathcal{N}(\mathbf{m}_Y, \mathbf{\Sigma}_Y)} W_p(\mu, \nu)$$
$$\text{s.t.} \quad \mathbf{m}_Y \in \mathbb{R}^D \tag{9}$$
$$\text{rank}(\mathbf{\Sigma}_Y) = \mathrm{d}$$

*is achieved when $\mathbf{m}_Y = \mathbf{m}_X$ and $\mathbf{\Sigma}_Y = \mathbf{P}_{\mathscr{L}} \mathbf{\Sigma}_X \mathbf{P}_{\mathscr{L}}$, where for $X \sim \mu$*

$$\mathscr{L} = \operatorname*{argmin}_{\dim \mathscr{L} = \mathrm{d}} \mathbb{E} \|X - \mathbf{P}_{\mathscr{L}} X\|_2^p. \tag{10}$$

The setting of this proposition implicitly assumes a linear generator of WGAN. Indeed, the linear mapping, which can be represented by a $d \times D$ matrix, maps a distribution in $\mathcal{N}(\mathbf{m}_X, \mathbf{\Sigma}_X)$ into a distribution in $\mathcal{N}(\mathbf{m}_Y, \mathbf{\Sigma}_Y)$ and reduces the rank of the covariance matrix from $D$ to $d$. The proposition states that in this setting the underlying minimization is closely related to minimizing the loss function (3). Note that here $p \geq 1$, however, if one further corrupts the sample, then $p = 1$ is the suitable choice (Lerman & Maunu, 2018). This choice is also more appropriate for WGAN, since there is no $p$-WGAN for $p \neq 1$.

Nevertheless, training a WGAN is not exactly the same as minimizing the $W_1$ distance (Gulrajani et al., 2017), since it is difficult to impose the Lipschitz constraint for a neural network. Furthermore, in practice, the WGAN generator, which is a neural network, is nonlinear, and thus its output is typically non-Gaussian. The robustness of WGAN with a linear autoencoder, which we established here, does not extend to a general WGAN (this is similar to our earlier observation that the robustness of a linear autoencoder with an RSR loss does not generalize to a nonlinear autoencoder). We believe that a similar structure like the RSR layer has to be imposed for enhancing the robustness of WGAN, and possibly also other generative networks, but we leave its effective implementation as an open problem.

## 6  Conclusion and future work

We constructed a simple but effective RSR layer within the autoencoder structure for anomaly detection. It is easy to use and adapt. We have demonstrated competitive results for image and document data and believe that it can be useful in many other applications.

There are several directions for further exploration of the RSR loss in unsupervised deep learning models for anomaly detection. First, we are interested in theoretical guarantees for RSRAE. A more direct subproblem is understanding the geometric structure of the "manifold" learned by RSRAE. Second, it is possible that there are better geometric methods to robustly embed the manifold of inliers. For example, one may consider a multiscale incorporation of RSR layers, which we expand on in Appendix D.3. Third, one may try to incorporate an RSR layer in other neural networks for anomaly detection that use nonlinear dimension reduction. We hope that some of these methods may be easier to directly analyze than our proposed method. For example, we are curious about successful incorporation of robust metrics for GANs or WGANs. In particular, we wonder about extensions of the theory proposed here for WGAN when considering a more general setting.

### Acknowledgments

This research has been supported by NSF award DMS18-30418. Part of this work was pursued when Dongmian Zou was a postdoctoral associate at the Institute for Mathematics and its Applications at the University of Minnesota. We thank Teng Zhang for his help with proving Proposition 5.1 (we discussed a related but different proposition with similar ideas of proofs). We thank Madeline Handschy for commenting on an earlier version of this paper.

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

## A   DETAILS OF RSRAE AND RSRAE+

The implementations of both RSRAE and RSRAE+ are simple. For completeness we provide here their details in algorithm boxes. The codes will be later posted in a supplementary webpage. Algorithm 1 describes RSRAE, which minimizes (5) by alternating minimization. It denotes the vectors of parameters of the encoder and decoder by $\boldsymbol{\theta}$ and $\boldsymbol{\varphi}$, respectively.

---

**Algorithm 1** RSRAE

---

**Input:** Data $\{\mathbf{x}^{(t)}\}_{t=1}^N$; thresholds $\epsilon_{\text{AE}}$, $\epsilon_{\text{RSR}_1}$, $\epsilon_{\text{RSR}_2}$, $\epsilon_{\text{T}}$; architecture and initial parameters of
    $\mathscr{E}$, $\mathscr{D}$, $\mathbf{A}$ (including number of columns of $\mathbf{A}$); number of epochs & batches; learning rate for
    backpropagation; similarity measure
**Output:** Labels of data points as normal or anomalous
 1: **for** each epoch **do**
 2:    Divide input data into batches
 3:    **for** each batch **do**
 4:       **if** $L_{\text{AE}}^1(\boldsymbol{\theta}, \mathbf{A}, \boldsymbol{\varphi}) > \epsilon_{\text{AE}}$ **then**
 5:          Backpropagate $L_{\text{AE}}^1(\boldsymbol{\theta}, \mathbf{A}, \boldsymbol{\varphi})$ w.r.t. $\boldsymbol{\theta}, \mathbf{A}, \boldsymbol{\varphi}$ & update $\boldsymbol{\theta}, \mathbf{A}, \boldsymbol{\varphi}$
 6:       **end if**
 7:       **if** $L_{\text{RSR}_1}^1(\mathbf{A}) > \epsilon_{\text{RSR}_1}$ **then**
 8:          Backpropagate $L_{\text{RSR}_1}^1(\mathbf{A})$ w.r.t. $\mathbf{A}$ & update $\mathbf{A}$
 9:       **end if**
10:       **if** $L_{\text{RSR}_2}^1(\mathbf{A}) > \epsilon_{\text{RSR}_2}$ **then**
11:          Backpropagate $L_{\text{RSR}_2}^1(\mathbf{A})$ w.r.t. $\mathbf{A}$ & update $\mathbf{A}$
12:       **end if**
13:    **end for**
14: **end for**
15: **for** $t = 1, \ldots, N$ **do**
16:    Calculate similarity between $\mathbf{x}^{(t)}$ and $\tilde{\mathbf{x}}^{(t)}$
17:    **if** similarity $\geq \epsilon_{\text{T}}$ **then**
18:       $\mathbf{x}^{(t)}$ is normal
19:    **else**
20:       $\mathbf{x}^{(t)}$ is anomalous
21:    **end if**
22: **end for**
23: **return**  Normality labels for $t = 1, \ldots, N$

---

We clarify some guidelines for choosing default parameters, which we follow in all reported experiments. We set $\epsilon_{\text{AE}}$, $\epsilon_{\text{RSR}_1}$ and $\epsilon_{\text{RSR}_2}$ to be zero. In general, we use networks with dense layers but for image data we use convolutional layers. We prefer using tanh as the activation function due to its smoothness. However, for a dataset that does not lie in the unit cube, we use either a ReLU function if all of its coordinates are positive, or a leaky ReLU function otherwise. The network parameters and the elements of $\mathbf{A}$ are initialized to be i.i.d. standard normal. In all numerical experiments, we set the number of columns of $\mathbf{A}$ to be 10, that is, $d = 10$. The learning rate is chosen so that there is a sufficient improvement of the loss values after each epoch. Instead of fixing $\epsilon_{\text{T}}$, we report the AUC and AP scores for different values of $\epsilon_{\text{T}}$.

Algorithm 2 describes RSRAE+, which minimizes (5) with fixed $\lambda_1$ and $\lambda_2$ by auto-differentiation.

---

**Algorithm 2** RSRAE+

---

**Input:** Data $\{\mathbf{x}^{(t)}\}_{t=1}^{N}$; thresholds $\epsilon_{\mathrm{AE}}$, $\epsilon_{\mathrm{T}}$; architecture and initial parameters of $\mathscr{E}$, $\mathscr{D}$, $\mathbf{A}$ (including number of columns of $\mathbf{A}$); parameters of the the energy function $\lambda_1$, $\lambda_2$; number of epochs & batches; learning rate for backpropagation; similarity measure

**Output:** Labels of data points as normal or anomalous
1: **for** each epoch **do**
2:    Divide input data into batches
3:    **for** each batch **do**
4:       **if** $L_{\mathrm{AE}}^1(\boldsymbol{\theta}, \mathbf{A}, \boldsymbol{\varphi}) > \epsilon_{\mathrm{AE}}$ **then**
5:          Backpropagate $L_{\mathrm{AE}}^1(\boldsymbol{\theta}, \mathbf{A}, \boldsymbol{\varphi}) + \lambda_1 L_{\mathrm{RSR}_1}^1(\mathbf{A}) + \lambda_2 L_{\mathrm{RSR}_2}^1(\mathbf{A})$ w.r.t. $\boldsymbol{\theta}, \mathbf{A}, \boldsymbol{\varphi}$ & update $\boldsymbol{\theta}, \mathbf{A}, \boldsymbol{\varphi}$
6:       **end if**
7:    **end for**
8: **end for**
9: **for** $t = 1, \ldots, N$ **do**
10:    Calculate similarity between $\mathbf{x}^{(t)}$ and $\tilde{\mathbf{x}}^{(t)}$
11:    **if** similarity $\geq \epsilon_{\mathrm{T}}$ **then**
12:       $\mathbf{x}^{(t)}$ is normal
13:    **else**
14:       $\mathbf{x}^{(t)}$ is anomalous
15:    **end if**
16: **end for**
17: **return** Normality labels for $t = 1, \ldots, N$

---

## B  DEMONSTRATION OF RSRAE FOR ARTIFICIAL DATA

For illustrating the performance of RSRAE, in comparison with a regular autoencoder, we consider a simple artificial geometric example. We assume corrupted data whose normal part is embedded in a "Swiss roll manifold"[3], which is a two-dimensional manifold in $\mathbb{R}^3$. More precisely, the normal part is obtained by mapping 1,000 points uniformly sampled from the rectangle $[3\pi/2, 9\pi/2] \times [0, 21]$ into $\mathbb{R}^3$ by the function

$$(s, t) \mapsto (t\cos(t), s, t\sin(t)). \tag{11}$$

The anomalous part is obtained by i.i.d. sampling of 500 points from an isotropic Gaussian distribution in $\mathbb{R}^3$ with zero mean and standard deviation 2 in any direction. Fig. 4a illustrates such a sample, where the inliers are in black and the outliers are in blue. We remark that Fig 5a is identical.

We construct the RSRAE with the following structure. The encoder is composed of fully-connected layers of sizes (32, 64, 128). The decoder is composed of fully connected layers of sizes (128, 64, 32, 3). Each fully connected layer is activated by the leaky ReLU function with $\alpha = 0.2$. The intrinsic dimension for the RSR layer, that, is the number of columns of $\mathbf{A}$, is $d = 2$.

For comparison, we construct the regular autoencoder AE (see Section 4.3). Recall that both of them have the same architecture (including the linear map $\mathbf{A}$), but AE minimizes the $\ell_2$ loss function in (6) (with $p = 2$) without an additional RSR loss. We optimize both models with 10,000 epochs and a batch gradient descent using Adam (Kingma & Ba, 2014) with a learning rate of 0.01.

The reconstructed data ($\tilde{\mathbf{X}}$) using RSRAE and AE are plotted in Figs. 4d and 5d, respectively. We further demonstrate the output obtained by the encoder and the RSR layer. The output of the encoder, $\mathbf{Z} = \mathscr{E}(\mathbf{X})$, lies in $\mathbb{R}^{128}$. For visualization purposes we project it onto a $\mathbb{R}^3$ as follows. We first find two vectors that span the image of $\mathbf{A}$ and we add to it the "principal direction" of $\mathbf{Z}$ orthogonal to the span of $\mathbf{A}$. We project $\mathbf{Z}$ onto the span of these 3 vectors. Figs. 4b and 5b show these projections for RSRAE and AE, respectively. Figs. 4c and 5c demonstrate the respective mappings of $\mathbf{Z}$ by $\mathbf{A}$ during the RSR layer.

Figs. 4d and 5d imply that the set of reconstructed normal points in RSRAE seem to lie on the original manifold, whereas the reconstructed normal points by AE seem to only lie near, but often

---

[3]https://scikit-learn.org/stable/modules/generated/sklearn.datasets.make_swiss_roll.html

not on the Swiss roll manifold. More importantly, the anomalous points reconstructed by RSRAE seem to be sufficiently far from the set of original anomalous points, unlike the reconstructed points by AE. Therefore, RSRAE can better distinguish anomalies using the distance between the original and reconstructed points, where small values are obtained for normal points and large ones for anomalous ones. Fig. 6 demonstrates this claim. They plot the histograms of the distance between the original and reconstructed points when applying RSRAE and AE, where distances for normal and anomalous points are distinguished by color. Clearly, RSRAE distinguishes normal and anomalous data better than AE.

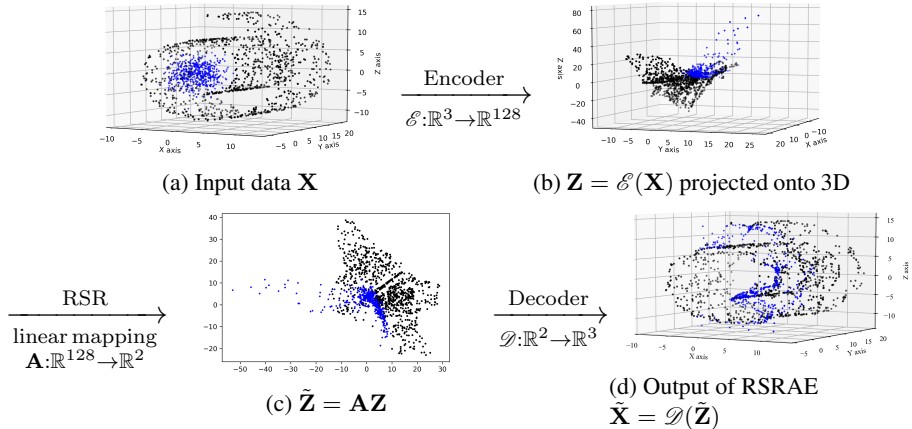

Figure 4: Demonstration of the output of the encoder, RSR layer and decoder of RSRAE on a corrupted Swiss roll dataset.

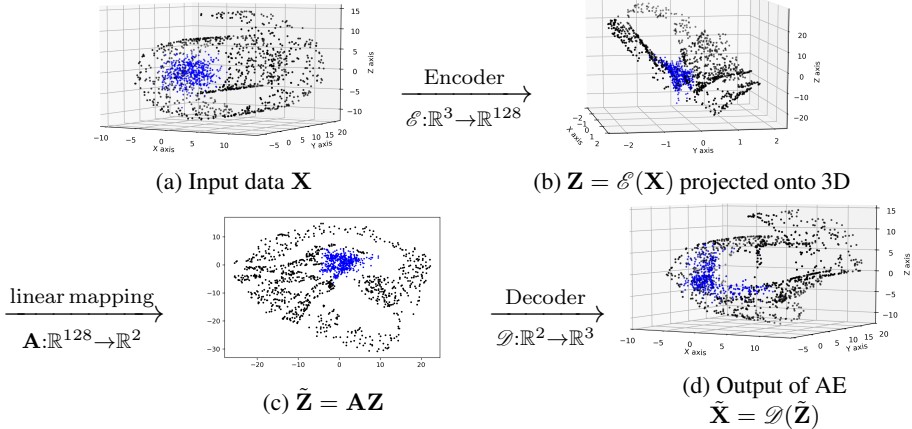

Figure 5: Demonstration of the output of the encoder, mapping by $\mathbf{A}$, and decoder of AE on a corrupted Swiss roll dataset.

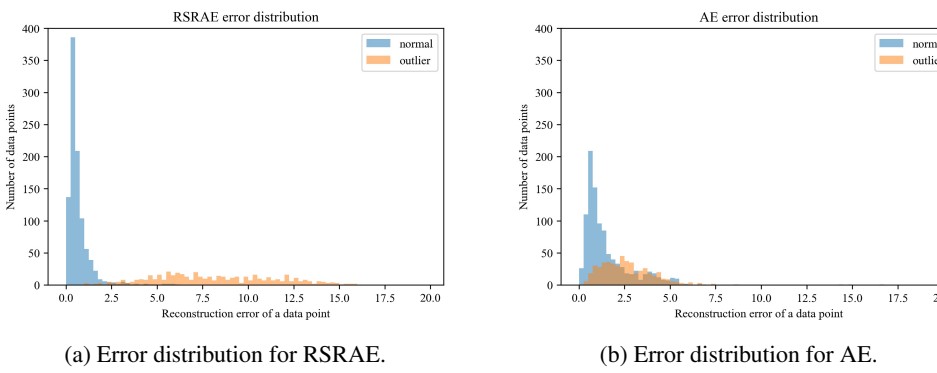

(a) Error distribution for RSRAE.

(b) Error distribution for AE.

Figure 6: Demonstration of the reconstruction error distribution for RSRAE and AE.

## C  FURTHER DISCUSSION OF THE RSR TERM

The RSR energy in (4) includes two different terms. The proposition below indicates that the second term of (4) is zero when plugging into it the solution of the minimization of the first term of (4) with the additional requirement that $\mathbf{A}$ has full rank. That is, in theory, one may only minimize the first term of (4) over the set of matrices $\mathbf{A} \in \mathbb{R}^{d \times D}$ with full rank. We then discuss computational issues of this different minimization.

**Proposition C.1.** *Assume that $\{\mathbf{z}^{(t)}\}_{t=1}^{N} \subset \mathbb{R}^D$ spans $\mathbb{R}^D$, $d \leqslant D$ and let*

$$\mathbf{A}^{\star} = \operatorname*{argmin}_{\substack{\mathbf{A} \in \mathbb{R}^{d \times D} \\ \mathrm{rank}(\mathbf{A})=\mathrm{d}}} \sum_{t=1}^{N} \left\| \mathbf{z}^{(t)} - \mathbf{A}^{\mathrm{T}} \mathbf{A} \mathbf{z}^{(t)} \right\|_2. \tag{12}$$

*Then $\mathbf{A}^{\star} \mathbf{A}^{\star \mathrm{T}} = \mathbf{I}_d$.*

*Proof.* Let $\mathbf{A}^{\star}$ be an optimizer of (12) and $\mathbf{P}^{\star}$ denote the orthogonal projection onto the range of $\mathbf{A}^{\star \mathrm{T}} \mathbf{A}^{\star}$. Note that $\mathbf{P}^{\star}$ can be written as $\tilde{\mathbf{A}}^{\mathrm{T}} \tilde{\mathbf{A}}$, where $\tilde{\mathbf{A}}$ is a $d \times D$ matrix composed of an orthonormal basis of the range of $\mathbf{P}^{\star}$. Therefore, being an optimum of (12), $\mathbf{A}^{\star}$ satisfies

$$\left\| \mathbf{z}^{(t)} - \mathbf{P}^{\star} \mathbf{z}^{(t)} \right\|_2 \geq \left\| \mathbf{z}^{(t)} - \mathbf{A}^{\star \mathrm{T}} \mathbf{A}^{\star} \mathbf{z}^{(t)} \right\|_2, \quad t = 1, \cdots, N. \tag{13}$$

On the other hand, the definition of orthogonal projection implies that

$$\left\| \mathbf{z}^{(t)} - \mathbf{P}^{\star} \mathbf{z}^{(t)} \right\|_2 \leq \left\| \mathbf{z}^{(t)} - \mathbf{A}^{\star \mathrm{T}} \mathbf{A}^{\star} \mathbf{z}^{(t)} \right\|_2, \quad t = 1, \cdots, N. \tag{14}$$

That is, equality is obtained in (13) and (14). This equality and the fact that $\mathbf{P}^{\star}$ is a projection on the range of $\mathbf{A}^{\star \mathrm{T}} \mathbf{A}^{\star}$ imply that

$$\mathbf{P}^{\star} \mathbf{z}^{(t)} = \mathbf{A}^{\star \mathrm{T}} \mathbf{A}^{\star} \mathbf{z}^{(t)}, \quad t = 1, \cdots, N. \tag{15}$$

Since $\{\mathbf{z}^{(t)}\}_{t=1}^{N}$ spans $\mathbb{R}^D$, (15) results in

$$\mathbf{P}^{\star} = \mathbf{A}^{\star \mathrm{T}} \mathbf{A}^{\star}, \tag{16}$$

which further implies that

$$\mathbf{A}^{\star} \mathbf{A}^{\star \mathrm{T}} \mathbf{A}^{\star} = \mathbf{A}^{\star} \mathbf{P}^{\star} = \mathbf{A}^{\star}. \tag{17}$$

Combining this observation $(\mathbf{A}^{\star} \mathbf{A}^{\star \mathrm{T}} \mathbf{A}^{\star} = \mathbf{A}^{\star})$ with the constraint that $\mathbf{A}^{\star}$ has a full rank, we conclude that $\mathbf{A}^{\star} \mathbf{A}^{\star \mathrm{T}} = \mathbf{I}_d$.

■

The minimization in (12) is nonconvex and intractable. Nevertheless, Lerman & Maunu (2017) propose a heuristic to solve it with some weak guarantees and Maunu et al. (2017) propose an algorithm with guarantees under some conditions. However, such a minimization is even more difficult when applied to the combined energy in (5), instead of (4). Therefore, we find it necessary to include the second term in (4) that imposes the nearness of $\mathbf{A}^{\mathrm{T}} \mathbf{A}$ to an orthogonal projection (equivalently, of $\mathbf{A} \mathbf{A}^{\mathrm{T}}$ to the identity).

# D    MORE ON RELATED THEORY FOR THE RSR PENALTY

In Section D.1 we characterize the solution of (7) via a subspace problem. Special case solutions to this problem include both the PCA subspace and the least absolute deviations subspace. In Section D.2 we prove Proposition 5.1. In Section D.3 we review some pure mathematical work that we find relevant to this discussion.

## D.1    PROPERTY OF LINEAR AUTOENCODERS

The following proposition expresses the solution of (7) in terms of another minimization problem. After proving it, we clarify that the other minimization problem is related to both PCA and RSR.

**Proposition D.1.** *Let* $p \geq 1$, $d < D$, *and* $\{\mathbf{x}^{(t)}\}_{t=1}^{N} \subset \mathbb{R}^D$ *be a dataset with rank at least* $d$. *If* $(\mathbf{D}^\star, \mathbf{E}^\star) \in \mathbb{R}^{D \times d} \times \mathbb{R}^{d \times D}$ *is a minimizer of (7), then*

$$\mathbf{D}^\star \mathbf{E}^\star = \mathbf{P}^\star \, , \tag{18}$$

*where* $\mathbf{P}^\star \in \mathbb{R}^{D \times D}$ *is a minimizer of*

$$\sum_{t=1}^{N} \left\| \mathbf{x}^{(t)} - \mathbf{P}\mathbf{x}^{(t)} \right\|_2^p \, , \tag{19}$$

*among all orthoprojectors* $\mathbf{P}$ *(that is,* $\mathbf{P} = \mathbf{P}^T$ *and* $\mathbf{P}^2 = \mathbf{P}$*) of rank* $d$.

*Proof.* Let $\mathbf{P}^\diamond$ be a minimizer of (19) and $(\mathbf{D}^\star, \mathbf{E}^\star)$ be a minimizer of (7). Since $\mathbf{P}^\diamond$ is an orthoprojector of rank $d$ it can be written as $\mathbf{P}^\diamond = \mathbf{U}^\diamond \mathbf{U}^{\diamond \mathrm{T}}$, where $\mathbf{U}^\diamond \in \mathbb{R}^{D \times d}$, and thus

$$\sum_{t=1}^{N} \left\| \mathbf{x}^{(t)} - \mathbf{D}^\star \mathbf{E}^\star \mathbf{x}^{(t)} \right\|_2^p \leq \sum_{t=1}^{N} \left\| \mathbf{x}^{(t)} - \mathbf{U}^\diamond \mathbf{U}^{\diamond \mathrm{T}} \mathbf{x}^{(t)} \right\|_2^p = \sum_{t=1}^{N} \left\| \mathbf{x}^{(t)} - \mathbf{P}^\diamond \mathbf{x}^{(t)} \right\|_2^p \, . \tag{20}$$

Let $\mathscr{L}$ denote the column space of $\mathbf{D}^\star \mathbf{E}^\star$. Then by the property of orthoprojection

$$\left\| \mathbf{x}^{(t)} - \mathbf{D}^\star \mathbf{E}^\star \mathbf{x}^{(t)} \right\|_2 \geq \left\| \mathbf{x}^{(t)} - \mathbf{P}_\mathscr{L} \mathbf{x}^{(t)} \right\|_2 \quad \text{for } 1 \leq t \leq N \tag{21}$$

and consequently

$$\sum_{t=1}^{N} \left\| \mathbf{x}^{(t)} - \mathbf{D}^\star \mathbf{E}^\star \mathbf{x}^{(t)} \right\|_2^p \geq \sum_{t=1}^{N} \left\| \mathbf{x}^{(t)} - \mathbf{P}_\mathscr{L} \mathbf{x}^{(t)} \right\|_2^p \geq \sum_{t=1}^{N} \left\| \mathbf{x}^{(t)} - \mathbf{P}^\diamond \mathbf{x}^{(t)} \right\|_2^p \, . \tag{22}$$

The combination of (20) and (22) yields the following two equalities

$$\sum_{t=1}^{N} \left\| \mathbf{x}^{(t)} - \mathbf{P}_\mathscr{L} \mathbf{x}^{(t)} \right\|_2^p = \sum_{t=1}^{N} \left\| \mathbf{x}^{(t)} - \mathbf{P}^\diamond \mathbf{x}^{(t)} \right\|_2^p \, , \tag{23}$$

$$\sum_{t=1}^{N} \left\| \mathbf{x}^{(t)} - \mathbf{D}^\star \mathbf{E}^\star \mathbf{x}^{(t)} \right\|_2^p = \sum_{t=1}^{N} \left\| \mathbf{x}^{(t)} - \mathbf{P}_\mathscr{L} \mathbf{x}^{(t)} \right\|_2^p \, . \tag{24}$$

We note that (23) implies that $\mathbf{P}_\mathscr{L}$ is a minimizer of (19) (among all rank $d$ orthoprojectors). We further note that (21) and (24) yield that for all $1 \leq t \leq N$

$$\left\| \mathbf{x}^{(t)} - \mathbf{D}^\star \mathbf{E}^\star \mathbf{x}^{(t)} \right\|_2 = \left\| \mathbf{x}^{(t)} - \mathbf{P}_\mathscr{L} \mathbf{x}^{(t)} \right\|_2 \, . \tag{25}$$

Since $\mathbf{D}^\star \mathbf{E}^\star \mathbf{x}^{(t)} \in \mathscr{L}$ and $\mathbf{P}_\mathscr{L}$ is an orthoprojector we conclude from (25) that

$$\mathbf{D}^\star \mathbf{E}^\star \mathbf{x}^{(t)} = \mathbf{P}_\mathscr{L} \mathbf{x}^{(t)} \quad \text{for } 1 \leq t \leq N. \tag{26}$$

We note that the definition of $(\mathbf{D}^\star, \mathbf{E}^\star)$ implies that $\mathscr{L}$ (which is the column space of $\mathbf{D}^\star \mathbf{E}^\star$) is contained in the span of $\{\mathbf{x}^{(t)}\}_{t=1}^{N}$. We also recall that the dimension of the span of $\{\mathbf{x}^{(t)}\}_{t=1}^{N}$ is at least the dimension of $\mathscr{L}$, that is, $d$. Combining the latter facts with (26) we obtain that $\mathbf{D}^\star \mathbf{E}^\star = \mathbf{P}_\mathscr{L}$. This and the fact that $\mathbf{P}_\mathscr{L}$ is a minimizer of (19) (which was derived from (23)) concludes (18). ∎

Note that when $p = 2$, the energy function in (19) corresponds to PCA. More precisely, a minimizer $\mathbf{P}^\star$ of (19) (among rank $d$ orthoprojectors) is an orthoprojector on a $d$-dimensional PCA subspace, equivalently, a subspace spanned by top $d$ eigenvectors of the sample covariance (we assume for simplicity linear, and not affine, autoencoder, so the PCA subspace is linear and thus when $p = 2$ the data is centered at the origin). This minimizer is unique if and only if the $d$-th eigenvalue of the sample covariance is larger than the $(d + 1)$-st eigenvalue. These elementary facts are reviewed in Section II-A of Lerman & Maunu (2018).

When $p = 1$, the minimizer $\mathbf{P}^\star$ of (19) (among rank $d$ orthoprojectors) is an orthoprojector on the $d$-dimensional least absolute deviations subspace. This subspace is reviewed in Section II-D of Lerman & Maunu (2018) as a common approach for RSR. The minimizer is often not unique, where sufficient and necessary conditions for local minima of (19) are studied in Lerman & Zhang (2014).

### D.2 Proof of Proposition 5.1

*Proof.* We denote the subspace $\mathscr{L}$ in the left hand side of (10) by $\mathscr{L}^\star$ in order to distinguish it from the generic notation $\mathscr{L}$ for subspaces. Consider the random variable $X \sim \mu$, Where $\mu$ is $\mathcal{N}(\mathbf{m}_X, \boldsymbol{\Sigma}_X)$. Fix $\pi \in \Pi(\mu, \nu)$. We note that

$$
\begin{aligned}
&\mathbb{E}_{(X,Y)\sim\pi} \|X - Y\|_2^p \\
&= \int_{\mathbb{R}^D} \int_{\mathbb{R}^D} \|\mathbf{x} - \mathbf{y}\|_2^p \, \pi(\mathbf{x}, \mathbf{y}) \mathrm{d}\mathbf{x} \, \mathrm{d}\mathbf{y} \\
&\geq \min_{\dim\mathscr{L}=d} \int_{\mathbb{R}^D} \mathrm{dist}(\mathbf{x}, \mathscr{L})^\mathrm{p} \int_{\mathbb{R}^D} \pi(\mathbf{x}, \mathbf{y}) \mathrm{d}\mathbf{y} \, \mathrm{d}\mathbf{x} \\
&= \min_{\dim\mathscr{L}=d} \int_{\mathbb{R}^D} \mathrm{dist}(\mathbf{x}, \mathscr{L})^\mathrm{p} \mu(\mathbf{x}) \, \mathrm{d}\mathbf{x} \\
&= \min_{\dim\mathscr{L}=d} \mathbb{E} \|X - \mathbf{P}_{\mathscr{L}} X\|_2^p \; .
\end{aligned}
\tag{27}
$$

The inequality in (27) holds since $X$ is fixed and $Y$ satisfies $(X, Y) \sim \pi$, so the distribution of $Y$ is $\mathcal{N}(\mathbf{m}_Y, \boldsymbol{\Sigma}_Y)$. Therefore, almost surely, $Y$ takes values in the $d$-dimensional affine subspace $\{\mathbf{y} \in \mathbb{R}^D : \mathbf{y} - \mathbf{m}_Y \in \mathrm{range}(\boldsymbol{\Sigma}_Y)\}$. Furthermore, we note that equality in (27) is achieved when $Y = \mathbf{P}_{\mathscr{L}^\star} X$.

We conclude the proof by showing that

$$
\mathbf{m}_X \in \mathscr{L}^\star. \tag{28}
$$

Indeed, (28) implies that the orthogonal projection of $X \sim \mathcal{N}(\mathbf{m}_X, \boldsymbol{\Sigma}_X)$ onto $\mathscr{L}^\star$ results in a random variable with distribution $\nu$ which is $\mathcal{N}(\mathbf{m}_X, \mathbf{P}_{\mathscr{L}^\star} \boldsymbol{\Sigma}_X \mathbf{P}_{\mathscr{L}^\star})$. By the above observation about the optimality of $Y = \mathbf{P}_{\mathscr{L}^\star} X$, the density of this distribution is the optimal solution of (9).

To prove (28), we assume without loss of generality that $\mathbf{m}_X = \mathbf{0}$. Denote the orthogonal projection of the origin onto the affine subspace $\mathscr{L}^\star$ by $\mathbf{m}_{\mathscr{L}^\star}$ and let $\mathscr{L}_0 = \mathscr{L}^\star - \mathbf{m}_{\mathscr{L}^\star}$. We need to show that $\mathscr{L}^\star = \mathscr{L}_0$, or equivalently, $\mathbf{m}_{\mathscr{L}^\star} = \mathbf{0}$. We note $\mathscr{L}_0$ is a linear subspace, $\mathbf{m}_{\mathscr{L}^\star}$ is orthogonal to $\mathscr{L}_0$ and thus there exists a rotation matrix $\mathbf{O}$ such that

$$
\mathbf{O}\mathscr{L}_0 = \{(0, \cdots, 0, z_{D-d+1}, \cdots, z_D) : z_{D-d+1}, \cdots z_D \in \mathbb{R}\} \; , \tag{29}
$$

and

$$
\mathbf{O}\mathbf{m}_{\mathscr{L}^\star} = (m_1, \cdots, m_{D-d}, 0, \cdots, 0) \; . \tag{30}
$$

For any $\mathbf{x} \in \mathbb{R}^D$ we note that $\mu(\mathbf{x}) = \mu(-\mathbf{x})$ since $\mu$ is Gaussian. Using this observation, other basic observations and the notation $\mathbf{O}\mathbf{x} = (x_1', \cdots, x_D')$ we obtain that

$$
\begin{aligned}
&\mathrm{dist}(\mathbf{x}, \mathscr{L}^\star)^\mathrm{p} \mu(\mathbf{x}) + \mathrm{dist}(-\mathbf{x}, \mathscr{L}^\star)^\mathrm{p} \mu(-\mathbf{x}) \\
&= \left(\mathrm{dist}(\mathbf{x}, \mathscr{L}^\star)^\mathrm{p} + \mathrm{dist}(-\mathbf{x}, \mathscr{L}^\star)^\mathrm{p}\right) \mu(\mathbf{x}) \\
&= \left(\mathrm{dist}(\mathbf{O}\mathbf{x}, \mathbf{O}\mathscr{L}^\star)^\mathrm{p} + \mathrm{dist}(-\mathbf{O}\mathbf{x}, \mathbf{O}\mathscr{L}^\star)^\mathrm{p}\right) \mu(\mathbf{x}) \\
&= \left(\left(\sum_{i=1}^{D-d} (x_i' - m_i)^2\right)^{p/2} + \left(\sum_{i=1}^{D-d} (-x_i' - m_i)^2\right)^{p/2}\right) \mu(\mathbf{x})
\end{aligned}
$$

$$
\begin{aligned}
&= \left( \left( \sum_{i=1}^{D-d} (x'_i - m_i)^2 \right)^{p/2} + \left( \sum_{i=1}^{D-d} (x'_i + m_i)^2 \right)^{p/2} \right) \mu(\mathbf{x}) \\
&\geq 2 \left( \sum_{i=1}^{D-d} {x'_i}^2 \right)^{p/2} \mu(\mathbf{x}) \\
&= 2 \operatorname{dist}(\mathbf{Ox}, \mathbf{O}\mathscr{L}_0)^{\mathrm{p}} \mu(\mathbf{x}) \\
&= 2 \operatorname{dist}(\mathbf{x}, \mathscr{L}_0)^{\mathrm{p}} \mu(\mathbf{x}) \\
&= \left( \operatorname{dist}(\mathbf{x}, \mathscr{L}_0)^{\mathrm{p}} + \operatorname{dist}(-\mathbf{x}, \mathscr{L}_0)^{\mathrm{p}} \right) \mu(\mathbf{x}) \\
&= \operatorname{dist}(\mathbf{x}, \mathscr{L}_0)^{\mathrm{p}} \mu(\mathbf{x}) + \operatorname{dist}(-\mathbf{x}, \mathscr{L}_0)^{\mathrm{p}} \mu(-\mathbf{x}) .
\end{aligned}
\tag{31}
$$

The inequality in (31) follows from the fact that for $p \geq 1$, the function $\|\cdot\|_2^p$ is convex as it is a composition of the convex function $\|\cdot\|_2 : \mathbb{R}^d \to \mathbb{R}_+$ and the increasing convex function $(\cdot)^p : \mathbb{R}_+ \to \mathbb{R}_+$. Equality is achieved in (31) if $m_i = 0$ for $i = 1, \cdots, D - d$, that is, $\mathscr{L}^\star = \mathscr{L}_0$.

Integrating the left and right hand sides of (31) over $\mathbb{R}^D$ results in

$$
\int_{\mathbb{R}^D} \operatorname{dist}(\mathbf{x}, \mathscr{L}^\star)^{\mathrm{p}} \mu(\mathbf{x}) \mathrm{d}\mathbf{x} \geq \int_{\mathbb{R}^D} \operatorname{dist}(\mathbf{x}, \mathscr{L}_0)^{\mathrm{p}} \mu(\mathbf{x}) \mathrm{d}\mathbf{x} .
\tag{32}
$$

Since $\mathscr{L}^\star$ is a minimizer among all affine subspaces of rank $d$ of $\int_{\mathbb{R}^D} \operatorname{dist}(\mathbf{x}, \mathscr{L})^{\mathrm{p}} \mu(\mathbf{x}) \, \mathrm{d}\mathbf{x} = \mathbb{E} \|X - \mathbf{P}_{\mathscr{L}} X\|_2^{\mathrm{p}}$, equality is obtained in (32). Consequently, equality is obtained, almost everywhere, in (31). Therefore, $\mathscr{L}^\star = \mathscr{L}_0$ and the claim is proved. ∎

### D.3 Relevant Mathematical Theory

We note that a complex network can represent a large class of functions. Consequently, for a sufficiently complex network, minimizing the loss function in (6) results in minimum value zero. In this case the minimizing "manifold" contains the original data, including the outliers. On the other hand, the RSR loss term imposes fitting a subspace that robustly fits only part of the data and thus cannot result in minimum value zero. Nevertheless, imposing a subspace constraint might be too restrictive, even in the latent space. A seminal work by Jones (1990) studies optimal types of curves that contain general sets. This work relates the construction and optimal properties of these curves with multiscale approximation of the underlying set by lines. It was generalized to higher dimensions in (**?**) and to a setting relevant to outliers in (**?**). These works suggest loss functions that incorporate several linear RSR layers from different scales. Nevertheless, their pure setting does not directly apply to our setting. We have also noticed various technical difficulties when trying to directly implement these ideas to our setting.

## E  Brief description of the baselines and metrics

We first clarify the methods used as baselines in Section 4.

**Local Outlier Factor (LOF)** measures the local deviation of a given data point with respect to its neighbors. If the LOF of a data point is too large then the point is determined to be an outlier.

**One-Class SVM (OCSVM)** learns a margin for a class of data. Since outliers contribute less than the normal class, it also applies to the unsupervised setting (Goldstein & Uchida, 2016). It is usually applied with a non-linear kernel.

**Isolation Forest (IF)** determines outliers by looking at the number of splittings needed for isolating a sample. It constructs random decision trees. A short path length for separating a data point implies a higher probability that the point is an outlier.

**Geometric Transformations (GT)** applies a variety of geometric transforms to input images and consequently creates a self-labeled dataset, where the labels are the types of transformations. Its anomaly detection is based on Dirichlet Normality score according to the softmax output from a classification network for the labels.

**Deep Structured Energy-Based Models (DSEBMs)** outputs an energy function which is the negative log probability that a sample follows the data distribution. The energy based model is connected to an autoencoder to avoid the need of complex sampling methods.

**Deep Autoencoding Gaussian Mixture Model (DAGMM)** is also a deep autoencoder model. It optimizes an end-to-end structure that contains both an autoencoder and an estimator for Gaussian Mixture Model. The anomaly detection is done after modeling the density function of the Gaussian Mixture Model.

Next, we review the definitions of the two metrics that we used: the AUC and AP scores (Davis & Goodrich, 2006). In computing these metrics we identify the outliers as "positive".

**AUC (area-under-curve)** is the area under the Receiver Operating Characteristic (ROC) curve. Recall that the True Positive Rate (TPR), or Recall, is the number of samples correctly labeled as positive divided by the total number of actual positive samples. The False Positive Rate (FPR), on the other hand, is the number of negative samples incorrectly labeled as positive divided by the total number of actual negative samples. The ROC curve is a graph of TPR as a function of FPR. It is drawn by recording values of FPR and TPR for different choices of $\epsilon_T$ in Algorithm 1.

**AP (average-precision)** is the area under the Precision-Recall Curve. While Recall is the TPR, Precision is the number of samples correctly labeled as positive divided by the total number of predicted positives. The Precision-Recall curve is the graph of Precision as a function of Recall. It is drawn by recording values of Precision and Recall for different choices of $\epsilon_T$ in Algorithm 1.

Both AUC and AP can be computed using the corresponding functions in the scikit-learn package (Pedregosa et al., 2011).

## F    COMPARISON WITH RSR AND RCAE

We demonstrate basic properties of our framework by comparing it to two different frameworks. The first framework is direct RSR, which tries to model the inliers by a low-dimensional subspace, as opposed to the nonlinear model discussed in here. Based on careful comparison of RSR methods in Lerman & Maunu (2018), we use the Fast Median Subspace (FMS) algorithm (Lerman & Maunu, 2017) and its normalized version, the Spherical FMS (SFMS). The other framework can be viewed a nonlinear version of RPCA, instead of RSR. It assumes sparse elementwise corruption of the data matrix, instead of corruption of whole data points, or equivalently, of some columns of the data matrix. For this purpose we use the Robust Convolutional Autoencoder (RCAE) algorithm of Chalapathy et al. (2017), who advocate it as "extension of robust PCA to allow for a nonlinear manifold that explains most of the data". We adopt the same network structures as in Section 4.1.

Fig. 7 reports comparisons of RSRAE, FMS, SFMS and RCAE on the datasets used in Section 4.2. We first note that both FMS and SFMS are not effective for the datasets we have been using. That is, the inliers in these datasets are not well-approximated by a linear model. It is also interesting to notice that without normalization to the sphere, FMS can be much worse than SFMS. That is, SFMS is often way more robust to outliers than FMS. This observation and the fact that there are no obvious normalization procedures a general autoencoder (see Section 5) clarifies why the mere use of the $L_{\mathrm{AE}}^1$ loss for an autoencoder is not expected to be robust enough to outliers.

Comparing with RSRAE, we note that RCAE is not a competitive method for these datasets. This is not surprising since the model of RCAE, which assumes sparse elementwise corruption, does not fit well to the problem of anomaly detection, but to other problems, such as background detection.

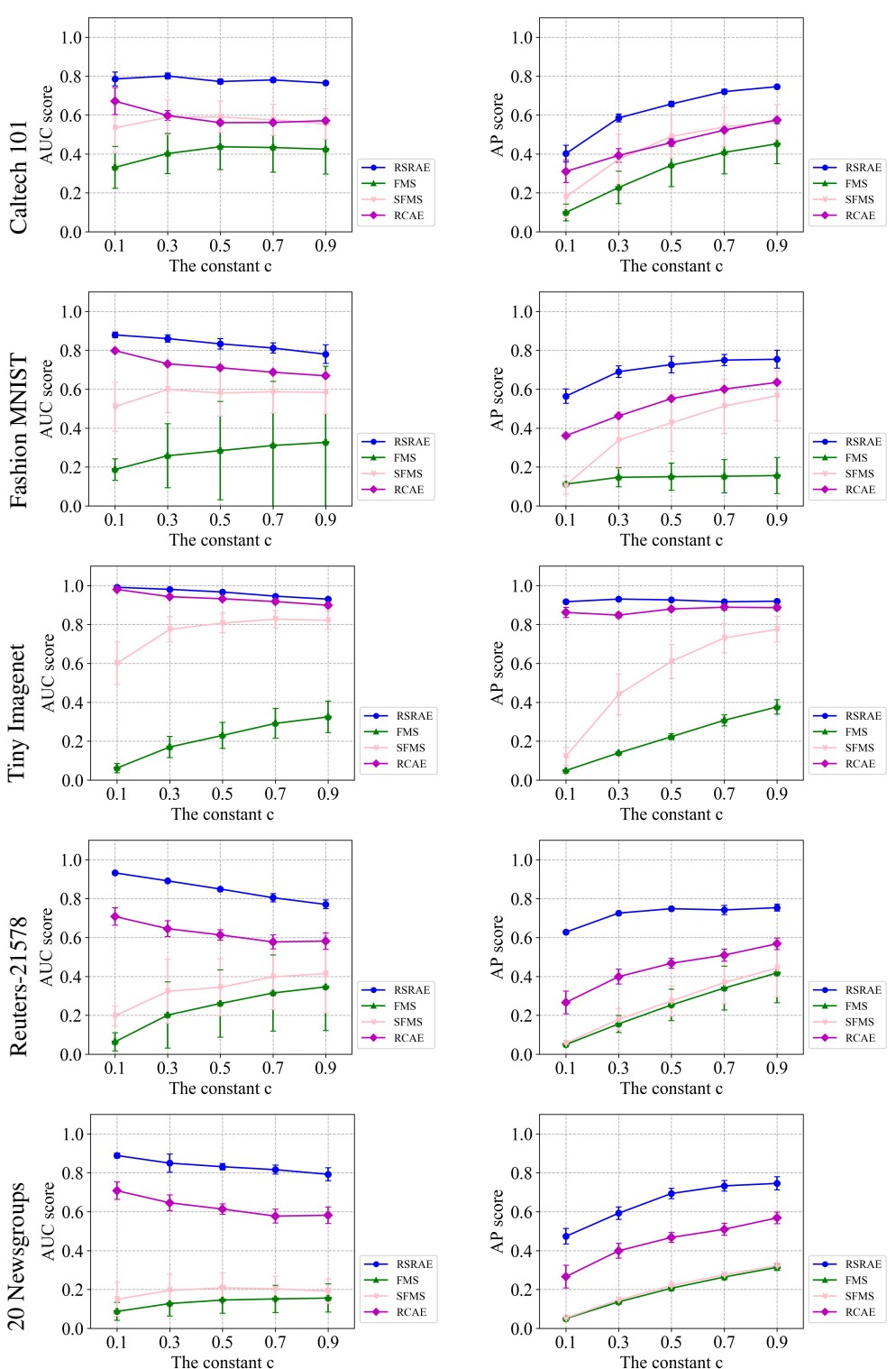

Figure 7: AUC and AP scores for RSRAE, FMS, SFMS and RCAE. From top to bottom are the results using Caltech 101, Fashion MNIST, Tiny Imagenet with deep features, Reuters-21578 and 20 Newsgroups.

# G   SENSITIVITY TO HYPERPARAMETERS

We examine the sensitivity of some of the reported results to changes in the hyperparameters. Section G.1 tests the sensitivity of RSRAE to changes in the intrinsic dimension $d$. Section G.2 tests the sensitivity of RSRAE to changes in the learning rate. Section G.3 tests the sensitivity of RSRAE+ to changes in $\lambda_1$ and $\lambda_2$.

## G.1   SENSITIVITY TO THE INTRINSIC DIMENSION

In the experiments reported in Section 4 we fixed $d = 10$. Here we check the sensitivity of the reported results to changes in $d$. We use the same datasets of Section 4.2 with an outlier ratio of $c = 0.5$ and test the following values of $d$: $1, 2, 5, 8, 10, 12, 15, 20, 30, 40, 50$. Fig. 8 reports the AUC and AP scores for these choice of $d$ and for these datasets with $c = 0.5$. We note that, in general, our results are not sensitive to choices of $d \le 30$.

We believe that the structure of these datasets is complex, and is not represented by a smooth manifold of a fixed dimension. Therefore, low-dimensional encoding of the inliers is beneficial with various choices of low dimensions.

When $d$ gets closer to $D$ the performance deteriorates. Such a decrease in accuracy is noticeable for Reuters-21578 and 20 Newsgroups, where for both datasets $D = 128$. For the image data sets (without deep features) $D = 1152$ and thus only relatively small values of $d$ were tested. As an example of large $d$ for an image dataset, we consider the case of $d = D = 1152$ in Caltech101 with $c = 0.5$. In this case, AUC = 0.619 and AP = 0.512, which are very low scores.

We conclude that in our experiments (with $c = 0.5$), RSRAE was stable in $d$ around our choice of $d = 10$.

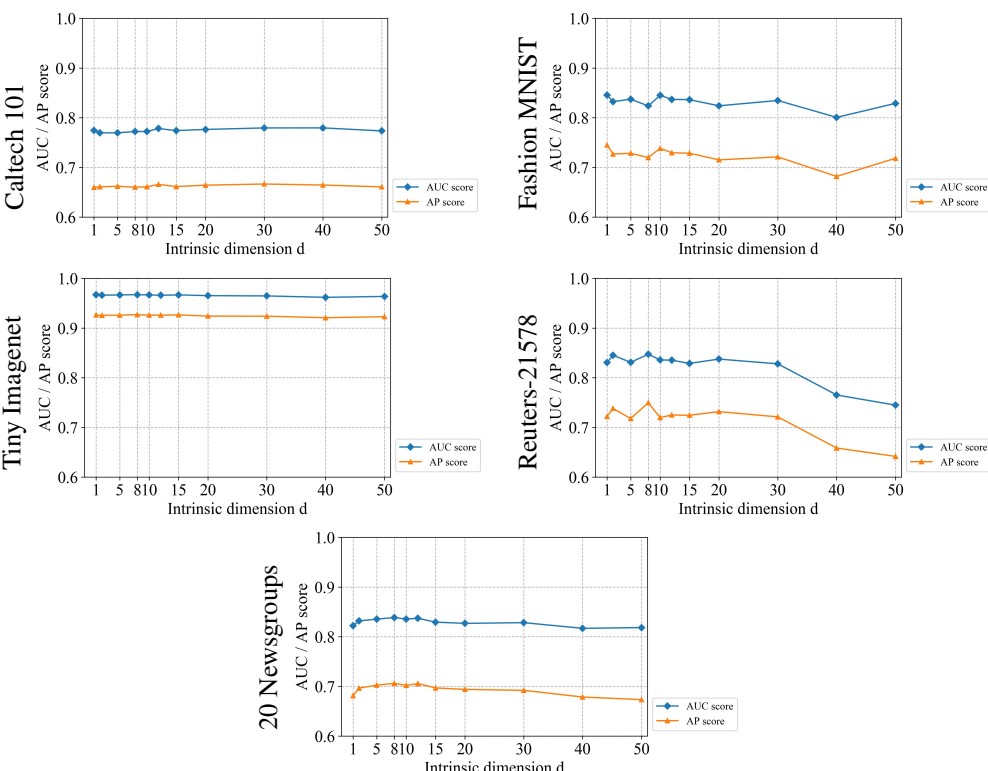

Figure 8: AUC and AP scores for different choices of $d$. The datasets are the same as those in Section 4.2, where the outlier ratio is $c = 0.5$.

## G.2 SENSITIVITY TO THE LEARNING RATE

In the experiments reported in Section 4 we fixed the learning rate for RSRAE to be $0.00025$. Here we check the sensitivity of the reported results to changes in the learning rate. We use the same datasets of Section 4.2 with an outlier ratio of $c = 0.5$ and test the following values of the learning rate: $0.0001, 0.00025, 0.0005, 0.001, 0.0025, 0.005, 0.01, 0.025, 0.05, 0.1$. Fig. 9 reports the AUC and AP scores for these values and for these datasets (with $c = 0.5$). We note that the performance is stable for learning rates not exceeding $0.01$.

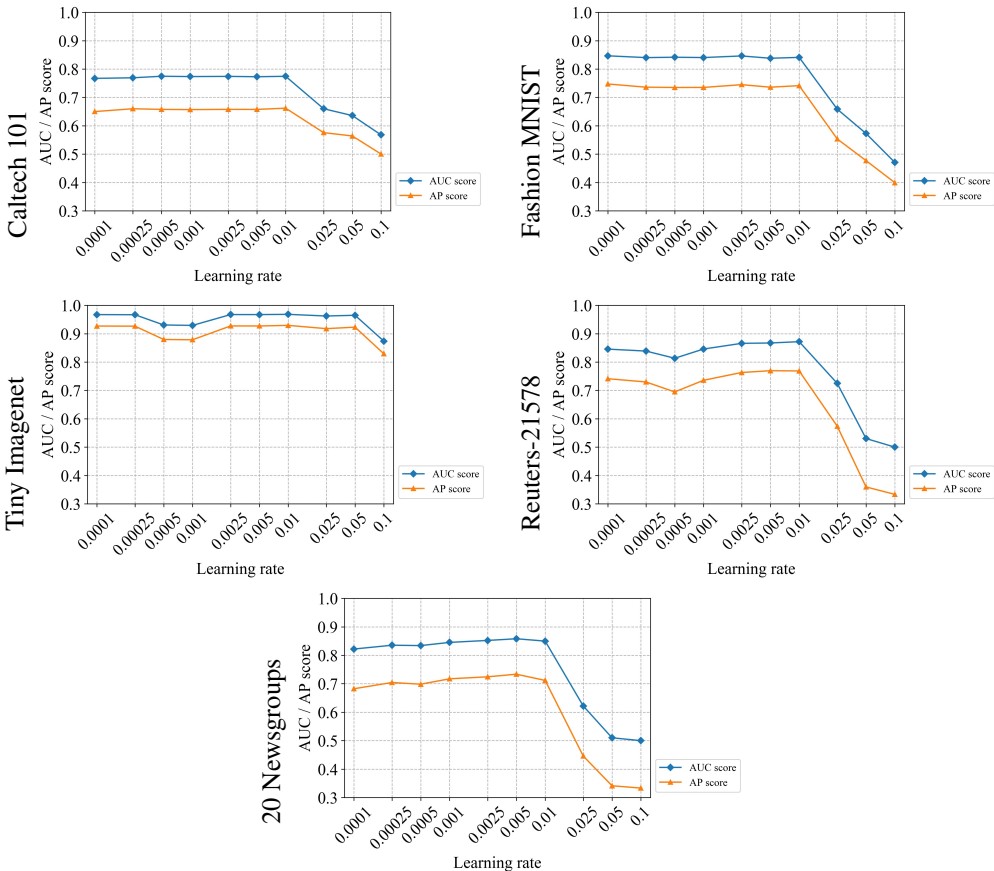

Figure 9: AUC and AP scores for various learning rates. The datasets are the same as those in Section 4.2, where the outlier ratio is $c = 0.5$.

## G.3 SENSITIVITY OF RSRAE+ TO $\lambda_1$ AND $\lambda_2$

We study the sensitivity of RSRAE+ to different choices of $\lambda_1$ and $\lambda_2$. We recall that RSRAE does not require these parameters. It is still interesting to check such sensitivity and find out whether careful tuning of these parameters in RSRAE+ can yield better scores than those of RSRAE. We use the same datasets of Section 4.2 with an outlier ratio of $c = 0.5$ and simultaneously test the following values of either $\lambda_1$ or $\lambda_2$: $0.01, 0.02, 0.05, 0.1, 0.2, 0.5, 1.0, 2.0$. Figs. 10 and 11 report the AUC and AP scores for these values and datasets (with $c = 0.5$). For each subfigure, the above values of $\lambda_1$ and $\lambda_2$ are recorded on the $x$ and $y$ axes, respectively. The darker colors of the heat map correspond to larger scores. For comparison, the corresponding AUC or AP score of RSRAE is indicated in the title of each subfigure.

We note that RSRAE+ is more sensitive to $\lambda_1$ than $\lambda_2$. Furthermore, as $\lambda_1$ increases the scores are often more stable to changes in $\lambda_1$. That is, the magnitudes of the derivatives of the scores with respect to $\lambda_1$ seem to generally decrease with $\lambda_1$. In Section 4.3 we used $\lambda_1 = \lambda_2 = 0.1$ as this choice seemed optimal for the independent set of 20 Newsgroup. We note though that optimal

hyperparameters depend on the dataset and it is thus not a good idea to optimize them using different datasets. They also depend on the choice of $c$, but for brevity we only test them with $c = 0.5$.

At last we note that the AUC and AP scores of RSRAE are comparable to the fine-tuned ones of RSRAE+ (where $c = 0.5$). We thus advocate using the alternating minimization of RSRAE, which is independent of $\lambda_1$ and $\lambda_2$.

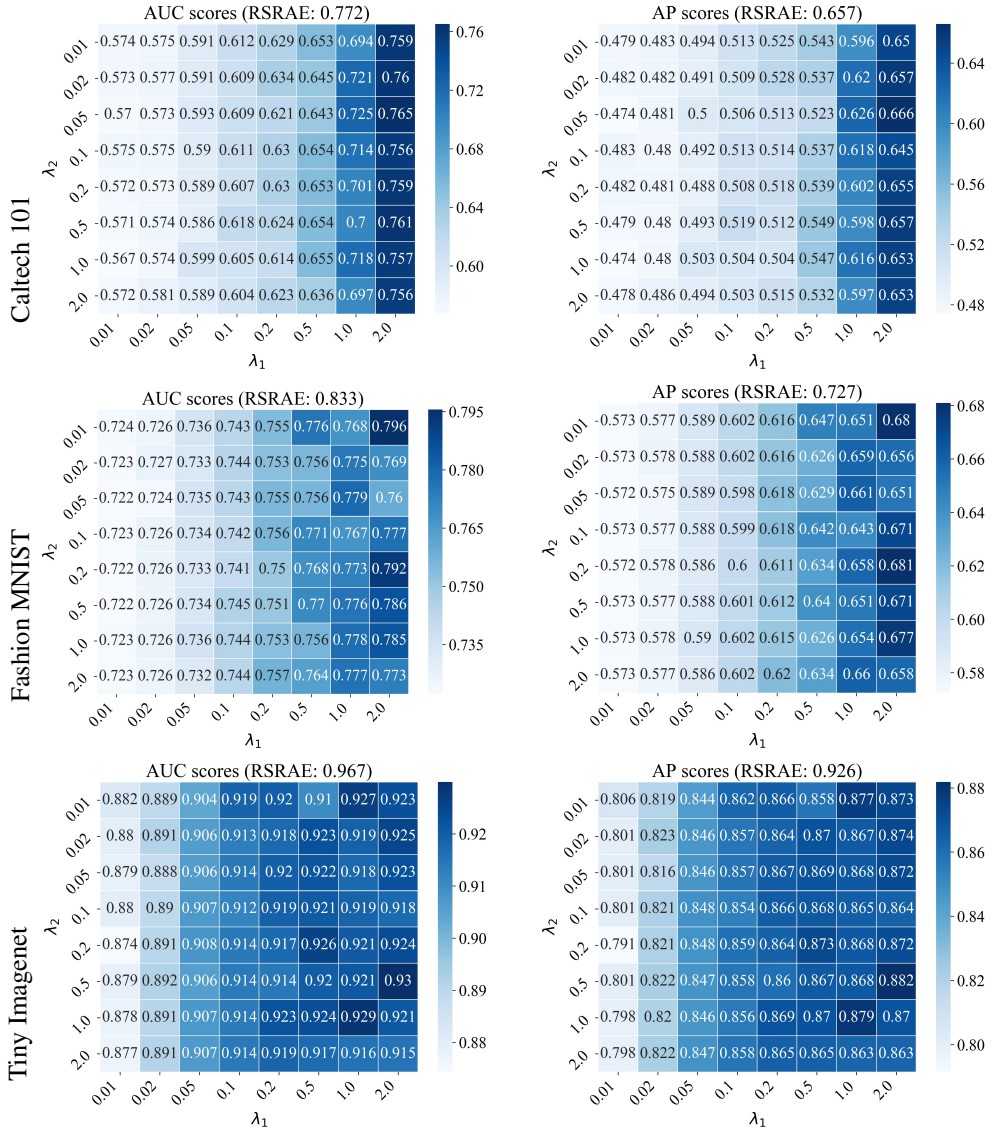

Figure 10: AUC and AP scores for RSRAE+ with various choices of $\lambda_1$ and $\lambda_2$ for Caltech 101, Fashion MNIST and Tiny Imagenet with deep features, where $c = 0.5$.

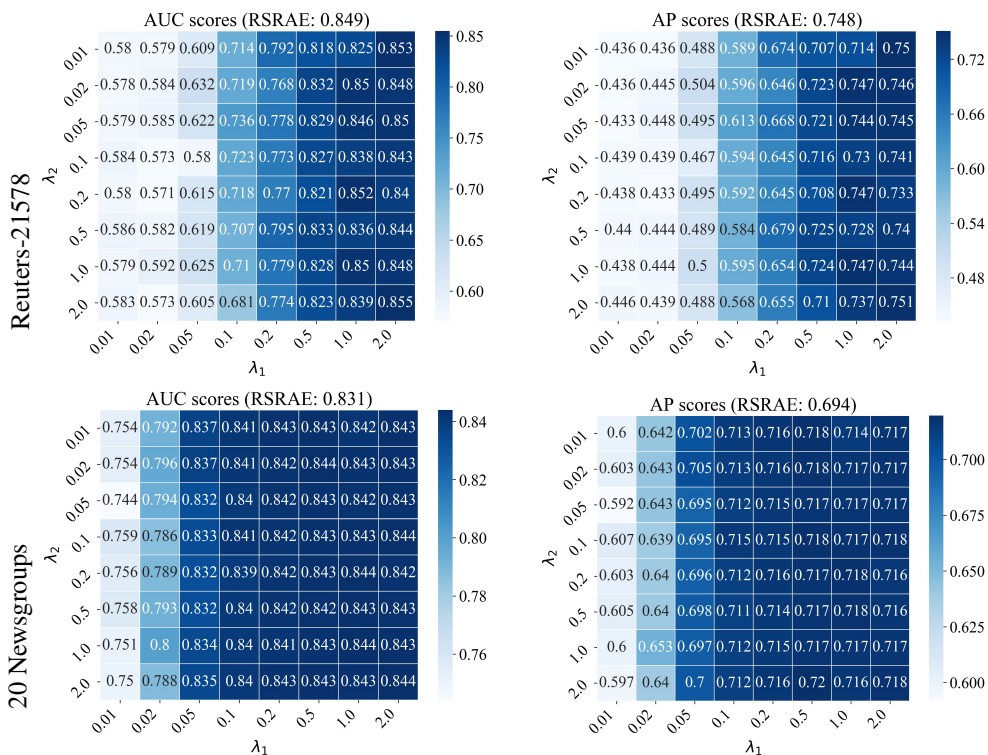

Figure 11: AUC and AP scores for RSRAE+ with various choices of $\lambda_1$ and $\lambda_2$ using Reuters-21578 and 20 Newsgroup, where $c = 0.5$.

# H  RUNTIME COMPARISON

Table 1 records runtimes for all the methods and datasets in Section 4.2 with the choice of $c = 0.5$. More precisely, a runtime is the the time needed to complete a single experiment, where 200 epochs were used for the neural networks. The table averages each runtime over the different classes.

Note that LOF, OCSVM and IF are faster than the rest of methods since they do not require training neural networks. We also note that the runtime of RSRAE is competitive in comparison to the other tested methods, that is, DSEBMs, DAGMM, and GT. The neural network structures of these four methods are the same, and thus the difference in runtime is mainly due to different pre and post processing.

Table 1: Runtime comparison: runtimes (in seconds) are reported for all methods and datasets in Section 4.2, where the outlier ratio is $c = 0.5$. Since GT was only applied to the image datasets without deep features, its runtime is not available (N/A) for the last three datasets.

| Benchmarks | Datasets Caltech 101 | Fashion MNIST | Tiny Imagenet | Reuters-21578 | 20 Newsgroups |
|---|---|---|---|---|---|
| LOF | 0.233 | 7.163 | 0.707 | 25.342 | 10.516 |
| OCSVM | 0.120 | 3.151 | 0.473 | 8.726 | 4.169 |
| IF | 0.339 | 1.485 | 0.511 | 20.481 | 6.751 |
| GT | 21.681 | 87.729 | N/A | N/A | N/A |
| DSEBMs | 14.293 | 46.933 | 25.194 | 41.083 | 33.852 |
| DAGMM | 21.066 | 71.632 | 41.211 | 83.551 | 60.720 |
| RSRAE | 6.305 | 33.853 | 10.940 | 32.061 | 18.869 |

## I ADDITIONAL RESULTS

We include some supplementary numerical results. In Section I.1 we show the results for Tiny Imagenet without deep features. In Section I.2 we extend the results reported in section 4.3 for the other datasets.

### I.1 TINY IMAGENET WITHOUT DEEP FEATURES

Fig. 12 presents the results for Tiny Imagenet without deep features. We see that RSRAE performs the best, but in general all the methods do not perform well. Indeed, the performance is significantly worse to that with deep features.

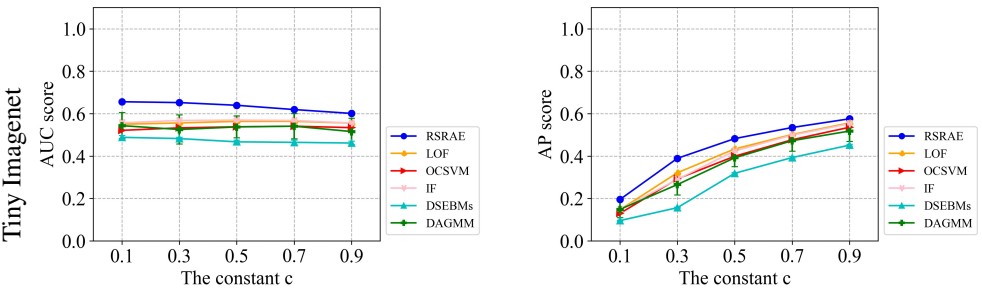

Figure 12: AUC and AP scores for the Tiny Imagenet without using the deep features.

### I.2 ADDITIONAL COMPARISON WITH VARIATIONS OF RSRAE

Figs. 13 and 14 extend the comparisons in Section 4.3 for additional datasets. The conclusion is the same. In general, RSRAE performs better by a large margin than AE and AE-1. On the other hand, RSRAE+ is often in between RSRAE and AE/AE-1. However, for 20 Newsgroups, RSRAE+ performs similarly to RSRAE, and possibly slightly better, than RSRAE. It seems that in this case our choice of $\lambda_1$ and $\lambda_2$ is good.

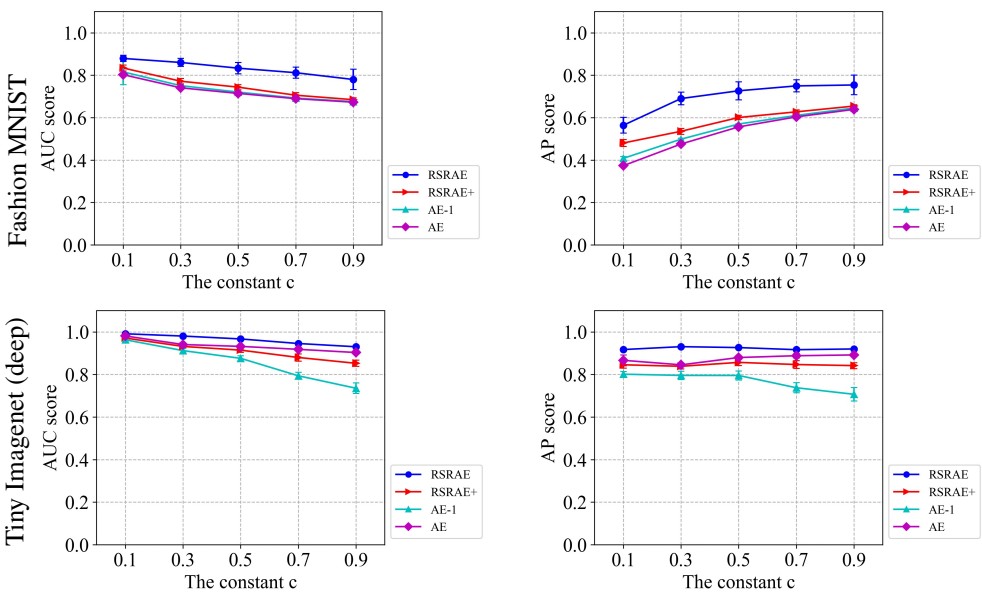

Figure 13: AUC and AP scores for RSRAE and alternative formulations using Fashion MNIST and deep features of Tiny Imagenet, where $c = 0.5$.

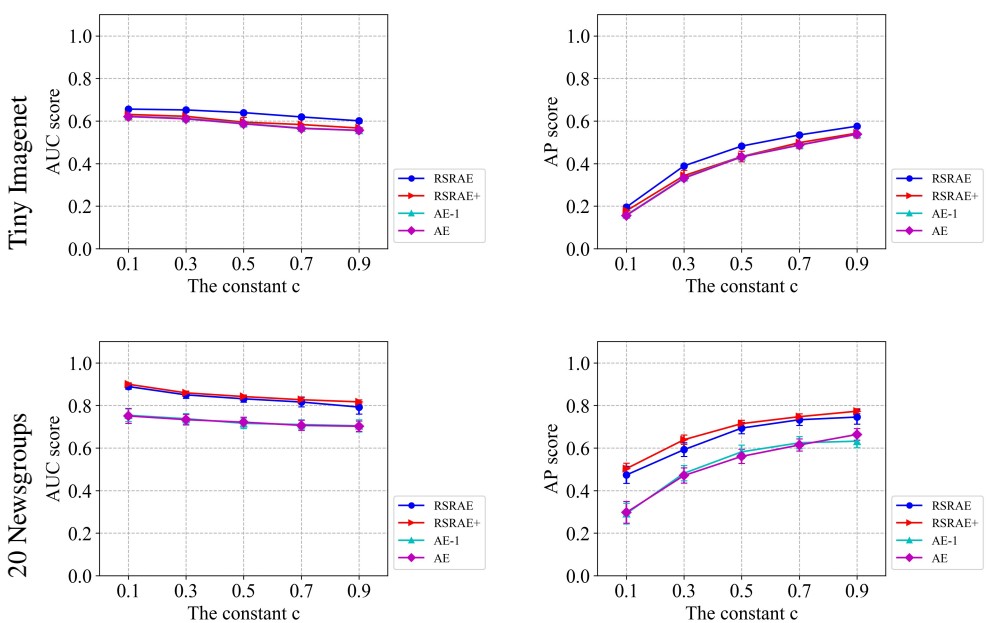

Figure 14: AUC and AP scores for RSRAE and alternative formulations using Tiny Imagenet (images) and 20 Newsgroup, where $c = 0.5$.

