# OpenReview forum: "Robust Subspace Recovery Layer for Unsupervised Anomaly Detection"
_ICLR.cc/2020/Conference — Accept (Poster)_

### Official Review · AnonReviewer3 · 2019-10-16
**Official Blind Review #3**

**Rating:** 8

**Review:**

This paper proposes to use the robust subspace recovery layer (RSR) in the autoencoder model for unsupervised anomaly detection.
This paper is well written overall. Presentation is clear and it is easy to follow.
The proposed approach is a simple combination of existing approaches.
Although its theoretical analysis with respect to the performance of anomaly detection is limited, experiments show that the proposed method is effective and superior to the existing anomaly detection methods.

I have the following comments:
- Parameter sensitivity should be examined.
    The proposed method has the number of parameters including \lambda_1, \lambda_2, and parameters in neural networks.
    Since parameter tuning is fundamentally difficult in the unsupervised setting, the sensitivity of the proposed method with respect to changes of such parameters should be examined.
- Since the efficiency is also an important issue for anomaly detection methods, runtime comparison would be interesting.
- It would be also interesting whether the proposed method is also effective for non-structured data, where a dataset is given as just a set of (real-valued) feature vectors, with its comparison to the standard anomaly detection methods such as LOF and iForest.


**Experience Assessment:**

I have published one or two papers in this area.

**Review Assessment: Checking Correctness Of Derivations And Theory:**

I assessed the sensibility of the derivations and theory.

**Review Assessment: Checking Correctness Of Experiments:**

I carefully checked the experiments.

**Review Assessment: Thoroughness In Paper Reading:**

I read the paper at least twice and used my best judgement in assessing the paper.

---

> ### Author Response · Authors · 2019-11-10
> **Response to Review #3**
>
> Thank you for your comments and constructive suggestions. We respond before the deadline, so we may communicate if needed.
>
> “
> The proposed approach is a simple combination of existing approaches.
> ”
> Response:  While AE and RSR are previous well-known approaches, RSRAE is not a simple combination of both of them. They are not two independent components that are combined together. In fact, we show that a simple combination of a least absolute deviations energy used in RSR and an AE, which we refer to AE-1, does not improve the performance of AE for anomaly detection. The RSR layer is between the encoder and decoder of the AE and it is optimized together with the autoencoder and not separately from it. We are planning to further clarify why this layer should be effective and what makes it unique.
>
> “
> - Parameter sensitivity should be examined.
>     The proposed method has the number of parameters including \lambda_1, \lambda_2, and parameters in neural networks.
>     Since parameter tuning is fundamentally difficult in the unsupervised setting, the sensitivity of the proposed method with respect to changes of such parameters should be examined.
> ”
> Response: We promoted the use of alternating minimization, which does not require predetermined $\lambda$’s. That is, the values of $\lambda_1$ and $\lambda_2$ are not relevant for the algorithm we advocated. They are only relevant to RSRAE+, but we do not advocate this algorithm due to the time it takes to test different values of lambda. Nevertheless, following this comment we will test the sensitivity of RSRAE+ to $\lambda_1$ and $\lambda_2$.
> We also agree that it is important to test the sensitivity with respect to the learning rate and the dimension of the subspace, and we will report it in the revised version.
>
> “
> - Since the efficiency is also an important issue for anomaly detection methods, runtime comparison would be interesting.
> ”
> Response: We plan to have a runtime comparison in the revised version.
>
> “
> - It would be also interesting whether the proposed method is also effective for non-structured data, where a dataset is given as just a set of (real-valued) feature vectors, with its comparison to the standard anomaly detection methods such as LOF and iForest.
> ”
> Response: When we tested our method on the deep features of Imagenet and the features of Reuters and 20 Newsgroups, we worked with a set of feature vectors. In those examples, we didn’t treat the features like images or sequences, and the network was simply a fully-connected network instead of a convolutional one. Please let us know if we missed anything.

---

### Official Review · AnonReviewer2 · 2019-10-21
**Official Blind Review #2**

**Rating:** 8

**Review:**

This paper adapts the concept of Robust Subspace Recovery (RSR) as a layer in an auto-encoder model for anomaly detection. A loss function is proposed that combines reconstruction error and a regularizer that enforces robustness against outliers. The reconstruction error expresses the accuracy of the nonlinear dimensionality reduction imposed by the autoencoder. The regularizer is the sum of absolute deviations from the latent subspace that represents a linear structure robust against outliers. An alternative procedure is applied where the loss terms are applied iteratively during training. Once trained, the reconstruction error is used directly for anomaly detection with a threshold. The AUC is used as a performance measure. The method is compared against 6 other methods (LOF, OCSVM, IF, DESBM, GT, DAGMM). The setting is fully unsupervised, meaning that the training data contains various amounts of anomalies, and the results are parametrized with the amount of corruption. The results show that the proposed approach outperforms the other methods in most cases, especially for larger amounts of corruption. An ablation study compares the approach with auto-encoder-only and a non-alternating gradient descent (fixed factors for each part of the loss function) and shows that the alternating method outperfroms all by a wide margin.


PROS:

* A novel approach to fully unsupervised anomaly detection that beats the state of the art.

* The RSR layer is a simple fully connected layer and the loss function is simple to calculate, making the approach computationally efficient.

* A pseudo-code algorithm is provided in the appendix, which should help reproducibility.

* The paper is well written and the math is clearly laid out.

* The result benchmarks are sufficiently exhaustive in both the methods that are compared and the datasets used.

* The ablation study is informative and shows the effect of the regularization term of the loss function as well as the effect of alternating the gradient descent with the separate losses.



CONS:

* There is a serious problem in the results (Figure 1) as the AP curves show better scores for larger corruption factors. Are the AP-score graphs flipped ? Please explain.

* The AUC and AP scores need to be defined.

* The results should include the case where the training data is not contaminated with outliers (c=0). This would correspond to the semi-supervised scenario and it would be very interesting to see how the method compares to DAGMM and GT which are build for that scenario.

* It would be interesting to see the effect of varying the subspace dimension. The authors chose 10 for all experiments, why is this number chosen, what would be the effect of choosing a smaller one ? This is a key parameter as it defines the structure of the projection subspace. Should this parameter be systematically tuned for each dataset ?


Overall this is a good paper proposing a novel approach to fully unsupervised anomaly detection with state-of-the art results.


**Experience Assessment:**

I have published one or two papers in this area.

**Review Assessment: Checking Correctness Of Derivations And Theory:**

I assessed the sensibility of the derivations and theory.

**Review Assessment: Checking Correctness Of Experiments:**

I carefully checked the experiments.

**Review Assessment: Thoroughness In Paper Reading:**

I read the paper thoroughly.

---

> ### Author Response · Authors · 2019-11-10
> **Response to Review #2**
>
> Thank you for your comments and constructive suggestions. We respond before the deadline, so we may communicate if needed.
>
> “
> * There is a serious problem in the results (Figure 1) as the AP curves show better scores for larger corruption factors. Are the AP-score graphs flipped ? Please explain.
> * The AUC and AP scores need to be defined.
> ”
> Response: The AUC and AP scores are calculated using the corresponding functions in the scikit-learn package. We will include the definitions in the appendix and will refer to scikit-learn.
>
> The AP scores are correct. They increase with c because the size of the outliers (that is, the size of the “Positive” class) increases with c. To clarify this issue, let’s assume a dataset with 2n points (where n > 1) with a single inlier and (2n-1) outliers and suppose an algorithm for anomaly detection that returns a similarity score (as mentioned in line 16 of Algorithm 1) of 0.5 for all inliers, 0.9 for (n-1) of the outliers and 0.1 for the other n outliers. Then AUC =  $n/(2n-1)$ while AP = $1 - (n-1)/(4n^2-2n)$. AUC decreases with n and AP increases with n.
>
>
> * The results should include the case where the training data is not contaminated with outliers (c=0). This would correspond to the semi-supervised scenario and it would be very interesting to see how the method compares to DAGMM and GT which are build for that scenario.
> ”
> Response: Our method was designed for the unsupervised setting. Indeed, RSRAE tries to extract the main structure from data in the presence of outliers without any information on the inliers. The main issue of the semi-supervised setting case is to learn the best model for the training data and use it for identifying outliers. If we use our method in a semi-supervised setting, then we only use its AE component without taking advantage of the RSR layer. This is not special to our method and similar results can be obtained by other autoencoders. Also, generative models may be advantageous for this problem.
>
> “
> * It would be interesting to see the effect of varying the subspace dimension. The authors chose 10 for all experiments, why is this number chosen, what would be the effect of choosing a smaller one? This is a key parameter as it defines the structure of the projection subspace. Should this parameter be systematically tuned for each dataset?
> ”
> Response: We plan to add some experiments addressing the subspace dimension. We chose 10 because we were interested in a single parameter that may work for a wide range of real datasets, and we generally noticed some stability to changes of this dimension. However, we agree that we need to report the experiments that demonstrate this stability.

---

### Official Review · AnonReviewer1 · 2019-10-24
**Official Blind Review #1**

**Rating:** 6

**Review:**

After reading all the reviews and the comments, I feel more positive about the paper. I appreciate the feedback of the Authors and I have decided to increase the rating.

============================

The paper proposes using Robust Subspace Recovery in combination with an autoencoder (and possibly GANs) for anomaly detection. The encoder maps input data to the latent space of dimensionality D, which then is linearly projected to a subspace of dimensionality d (d < D). The projection of the latent space then goes to a decoder that reconstructs the input.
A transformation matrix A is trained jointly with the autoencoder. Two additional terms are added to the loss: one to encourage the subspace of A^TA to approximate the latent space z and the second one to force it to be an orthogonal projector.

The paper claims to generalize the existing RSR framework to the nonlinear case. However, the linear RSR is applied to the latent space of the autoencoder. In addition to that, all the following discussion and proofs are limited to the linear case.

Since the proposed method is using RSB as it’s core part, and claims to be a non-linear extension of it, it would be crucial to have a comparison with RSB, at least on those experimental setups, where high-level features are used (Tiny Imagenet with ResNET features, Reuters-21578, and 20 Newsgroups). However, there is no such comparison.

Since autoencoders can potentially learn any, arbitrary entangled latent space, it is not clear why outliers should necessarily have such embedding that is outside of the learned subspace. In the case of the original RSR it happens due to the dimensionality reduction by the orthogonal projector. However, autoencoders already perform dimensionality reduction at each layer down to the bottleneck layer.

The matrix A and the parameters of the AE are trained jointly. So, it can be seen that two processes can occur:
- The AE in order to minimize the reconstruction error would learn such latent space z, that would fit into the subspace of A^TA, so that projection \tilde z =Az doesn’t cause data loss.
- The AE in order to minimize the reconstruction error would learn such A, so that the subspace that z approximates is the best possible.

It is not clear, which of the two cases would take place. If the first one would dominate, then it is not clear if such method would have any discriminating capabilities.

My point is mainly that the presented work is not really a generalization of RSR as it claims to be, but rather it is just using RSR on a leaned embedding of the data.

Some citations are missing, as well as it is missing a comparison to some state-of-the art methods such as OCNN ‘Robust, Deep and Inductive Anomaly Detection’ ECML 2017;   ‘Adversarially Learned One-Class Classifier for Novelty Detection’ CVPR 2017; DSVDD ‘Deep one-class classification.’ ICML, 2018; ODIN  ‘Enhancing The Reliability  Of Out-of-distribution Image Detection  In Neural Networks’ ICLR 2018; ‘Generative Probabilistic Novelty Detection with Adversarial Autoencoders’ NeurIPS 2018.


**Experience Assessment:**

I have published one or two papers in this area.

**Review Assessment: Checking Correctness Of Derivations And Theory:**

I carefully checked the derivations and theory.

**Review Assessment: Checking Correctness Of Experiments:**

I carefully checked the experiments.

**Review Assessment: Thoroughness In Paper Reading:**

I read the paper at least twice and used my best judgement in assessing the paper.

---

> ### Author Response · Authors · 2019-11-10
> **Response to Review #1 (part 1)**
>
> Thank you for your comments. We respond before the deadline, so we may communicate if needed.
>
> “The paper claims to generalize the existing RSR framework to the nonlinear case.”
> Response: Our paper does not claim to generalize the RSR framework to a nonlinear one, but it combines ideas from RSR within an autoencoder in order to make the autoencoder more robust to outliers.  The mere use of a “robust metric” (such as the least absolute deviations) for an autoencoder is not sufficiently robust to anomalies and this is why we find an idea like ours natural.  We believe that the above wrong interpretation of our paper is due to the following two sentences in Section 5: “Goodfellow et al. (2016) exemplified how PCA can be structured as a linear autoencoder. Similarly, RSR can be directly used to form an outliers-robust linear autoencoder and our current work generalizes this basic idea to a nonlinear setting”. The first claim here is that similarly to having an autoencoder for PCA (with linear encoder and decoder and least squares minimization), one can have an autoencoder for RSR (with linear encoder and decoder and absolute deviations minimization). Note that the difference between a PCA autoencoder and an RSR autoencoder is obtained by changing the minimization from least squares to absolute deviations (where it is also natural to introduce a simple normalization in the implementation part). On the other hand, changing the metric of a general autoencoder, which corresponds to “nonlinear PCA”, from least squares to absolute deviations does not make it robust (especially as ideas of simple normalizations for adversarial outliers do not work out in this case). We demonstrated this in our experiments (see performance of AE-1 vs. RSRAE in Section 4.3 and Appendix F2). We find that the RSR layer is a very natural idea to make an autoencoder robust. By writing “generalizes this basic idea to a nonlinear setting”, we meant extending the autoencoder to be robust to outliers, in analogy to making a PCA autoencoder robust by changing it into an RSR (linear) autoencoder, but we did not mean using the same method as in the latter one and formally generalizing RSR to the nonlinear case.
>
> The purpose of these two sentences was only to motivate the rigorous theory for WGAN. We considered them as part of a minor comment, which the expert can easily figure out. There was no claim about generalizing RSR to the nonlinear case in the introduction or the conclusion.  Due to the confusion, we will extend and rewrite this part with a careful explanation how our method is expected to make an autoencoder more robust to outliers, and why the mere change of a metric is not sufficient to make the method robust. Anyway, we don’t find these new details necessary for understanding the paper, but they may avoid a similar confusion.
>
> “However, the linear RSR is applied to the latent space of the autoencoder.”
>  Response: The linear layer is in the middle of an autoencoder and an RSR loss is part of the total loss function. It is not correct to think of two separate entities: an autoencoder and a linear RSR applied to the latent space of the autoencoder. That is, we do not try to first get the latent code form AE and then apply RSR to the latent code. Note that a general autoencoder tries to parametrize the structure of the data with a latent code within a low-dimensional linear space. Here we try to parametrize only the structure of the inliers, where the low-dimensional subspace lies within the ambient space of the encoder, we then map the output of the encoder onto a low-dimensional space corresponding to the latter subspace. This mapping is natural to the inliers, but not to the outliers, which are expected to have high reconstruction error.

---

> ### Author Response · Authors · 2019-11-10
> **Response to Review #1 (part 2)**
>
> “In addition to that, all the following discussion and proofs are limited to the linear case.”
> Response: Our discussion and experiments are for the nonlinear case. The fact that the transformation A is linear does not make the whole method linear. Let us also clarify that this is not a theoretical paper (even though it has some theory). It proposes a method and numerically tests it. We never claimed to provide theoretical justification for the proposed method. It does not seem feasible to theoretically justify this nonlinear procedure. It is also unfair to ask for theoretical justification as the effectiveness of the nonlinear dimension reduction by autoencoders is not well justified even though autoencoders have been around since the 80s and are now well recognized as effective practical tools for nonlinear dimension reduction.
> The proof in Appendix C does is not “limited to the linear case”, but it addresses the fact that the second term in (4) may be dropped in case one can find the minimizer in (10). We would like to emphasize the motivation of formulating and proving Proposition 5.1, which deals with a linear setting. Since the linear case of PCA motivates the notion of an autoencoder, we find it interesting to observe the case of a linear generator in WGAN and carefully understand its output, in particular, notice its relationship with the robust energy that we use. While we find the proofs of these propositions interesting (especially of the second proposition), they are not meant to address the performance of the main algorithm that we focus on and we believe that we made it clear in our writing.
>
> “Since the proposed method is using RSB as its core part, and claims to be a non-linear extension of it, it would be crucial to have a comparison with RSB, at least on those experimental setups, where high-level features are used (Tiny Imagenet with ResNet features, Reuters-21578, and 20 Newsgroups).”
> Response: Again, we emphasize that our method is not a direct extension of RSR to the nonlinear case and we do not claim so. Also, the RSR component is integrated in a very special way within the autoencoder and is not a separate component.
>
> More importantly, one cannot expect a linear model for the normal points to do well in practice and we also mentioned it in the paper. In particular, when considering a dataset of features, “high-level” features are not expected to lie in a common subspace. We can exemplify the poor performance of direct RSR on a particular dataset in the appendix, even though we believe it should be clear. We also want to clarify that what we call RSR is different than what is commonly referred to as RPCA, which the reviewer alludes to later.
>
>  “Since autoencoders can potentially learn any, arbitrary entangled latent space, it is not clear why outliers should necessarily have such embedding that is outside of the learned subspace. In the case of the original RSR it happens due to the dimensionality reduction by the orthogonal projector. However, autoencoders already perform dimensionality reduction at each layer down to the bottleneck layer.”
> Response: We don’t agree that “autoencoders can potentially learn any, arbitrary entangled latent space”.  The number of neurons in the autoencoder (and in particular, the dimension of the latent code) imposes a serious constraint.  Actually, you mentioned that there is a bottleneck layer in a regular autoencoder. Due to this bottleneck layer, it is impossible to learn an autoencoder that produces zero error for all the data points. We try to focus only on the inliers and have a latent subspace for the inliers within the ambient space of the encoder. If outliers are mapped to this subspace, as you may worry, then these outliers will have huge reconstruction errors, which will easily help distinguish them and thus RSRAE should perform well in this case. That is, the answer to your point “it is not clear why outliers should necessarily have such embedding that is outside of the learned subspace”,  is that there is no reason for the outliers to be outside the learned inliers’ subspace and it will be actually easier to distinguish them if they also happen to lie in this subspace. However, we wish to carefully learn such a subspace, while putting special emphasis on the inliers, instead of mapping all points to a low-dimensional subspace without recognizing properties of the inliers.  The loss function of RSRAE does not care much about the outliers, but takes careful care of the inliers. It tries to carefully learn their structure and use it to distinguish between inliers and outliers.

---

> ### Author Response · Authors · 2019-11-10
> **Response to Review #1 (part 3)**
>
> “The matrix A and the parameters of the AE are trained jointly. So, it can be seen that two processes can occur:
> - The AE in order to minimize the reconstruction error would learn such latent space z, that would fit into the subspace of A^TA, so that projection \tilde z =Az doesn’t cause data loss.
> - The AE in order to minimize the reconstruction error would learn such A, so that the subspace that z approximates is the best possible.
> It is not clear, which of the two cases would take place. If the first one would dominate, then it is not clear if such method would have any discriminating capabilities.
> ”
> Response: Our minimization does not wait until the convergence of the first term in eq. (5) of our paper before backpropagating the second and third terms in this equation. Instead, we alternatively backpropagate the three terms (please review the detailed pseudocode that we provided in the appendix). Therefore, it is unlikely that the algorithm only focuses on learning the best z that fits A. Nevertheless, even if the latent space for z is already low-rank before applying A, the method should still have discriminating capabilities because the latent space should correspond to the inliers. But this should not be the case in general, since otherwise, AE-1 would perform well (as opposed to the demonstrated results in Section 4.3 and Appendix F.2).
> The description above of the reviewer corresponds to that in Section 3.2 of  “Robust, Deep and Inductive Anomaly Detection” by R. Chalapathy, A. K. Menon and S. Chawla, but our paper is completely different than this paper.
>
> “My point is mainly that the presented work is not really a generalization of RSR as it claims to be, but rather it is just using RSR on a leaned embedding of the data. ”
> Response: Again, we did not advocate for establishing a generalization of RSR. We proposed a method that effectively addresses unsupervised anomaly detection by incorporating ideas of RSR within an autoencoder. Also, it is not true to claim that it uses RSR on the learned embedding of the data, as we explained above. The RSR loss plays an essential role in the learning process.
>
> “Some citations are missing, as well as it is missing a comparison to some state-of-the art methods such as OCNN ‘Robust, Deep and Inductive Anomaly Detection’ ECML 2017;   ‘Adversarially Learned One-Class Classifier for Novelty Detection’ CVPR 2017;  ‘Deep one-class classification.’ ICML, 2018; ODIN  ‘Enhancing The Reliability  Of Out-of-distribution Image Detection  In Neural Networks’ ICLR 2018; ‘Generative Probabilistic Novelty Detection with Adversarial Autoencoders’ NeurIPS 2018.”
> Response:  RCAE in “Robust, Deep and Inductive Anomaly Detection” (ECML, 2017) is indeed another unsupervised method for anomaly detection. However, RCAE is related to RPCA, where one assumes entry-wise corruption of the data matrix. On the other hand, we discuss here the case where whole data points are outliers (that is, rows of the data matrix are corrupted), instead of few entries of a matrix are corrupted. Therefore, we don’t expect that RCAE will work well for our model of outliers. We can make few comparisons with RCAE in the appendix or supplemental material.
>
> All the other methods you mentioned above are state-of-the-art for the semi-supervised setting (novelty detection), but here we emphasize a completely unsupervised setting (anomaly detection). Note that some of those methods (the ICLR 2018 and CVPR 2017 papers you mentioned) claim to be unsupervised anomaly detection / outlier detection but they are actually performing semi-supervised tasks, where one trains a model for the inliers. We already included GT from NIPS 2018 and DAGMM from ICLR 2018 and we do not find a reason to add more baseline methods, which are not unsupervised.
>
> As for the claim that “some citations are missing”, we will be happy to cite whatever relevant references we missed. Please let us know specific relevant papers that we failed to cite.

---

### Author Response · Authors · 2019-11-15
**Response to all reviewers with summary of changes**

Thanks again for the comments and suggestions.

We have modified our manuscript in order to address the concerns we responded to on November 10. The changes in the paper are indicated in red.

The major revisions are as follows:

- We clarified our contribution in Section 2.2. In particular, we distinguished our method from a direct application of RSR to a regular autoencoder (in response to R1 and R3). We also emphasized that our method is designed to handle the unsupervised setting and not the semi-supervised or supervised ones (in response to R2).

- We added the definition of AUC and AP in Appendix E and mentioned it in Section 4.2 (addressing R2).

- We added Section 5.1 in order to provide more intuition for RSRAE. We explained why an RSR layer is needed instead of a direct application of an $l_{2,1}$ within a regular autoencoder. A proposition for linear autoencoders with different loss functions (that include the cases of PCA and RSR) is proved in Appendix D.1. (Section 5.1 is motivated by various claims of R1, which we disagreed with in our earlier response. It tries to further clarify our work and avoid some misunderstandings).

- We added Appendix F, which compares RSRAE with (1) RSR (using both the FMS and SFMS algorithms) and (2) RCAE by Chalapathy, Menon and Chawla (2017).  (This addressed a request by R1).

- We added Appendix G, which includes sensitivity analysis. Section G.1 addresses sensitivity to the intrinsic dimension (addressing R2 and R3). Section G.2 addresses sensitivity to the learning rate (addressing R3) and Section G.3 addresses sensitivity of RSRAE+ to the lambda’s (addressing R3).

- We added runtime comparison in Appendix H (addressing R3).

---

### Decision · Program_Chairs · 2019-12-19

**Decision:**

Accept (Poster)

**Comment:**

Three reviewers have assessed this paper and they have scored it as 6/6/6/6 after rebuttal. Nonetheless, the reviewers have raised a number of criticisms and the authors are encouraged to resolve them for the camera-ready submission.